# Breath Analysis: A Promising Tool for Disease Diagnosis—The Role of Sensors

**DOI:** 10.3390/s22031238

**Published:** 2022-02-06

**Authors:** Maria Kaloumenou, Evangelos Skotadis, Nefeli Lagopati, Efstathios Efstathopoulos, Dimitris Tsoukalas

**Affiliations:** 1Department of Applied Physics, National Technical University of Athens, 15780 Athens, Greece; mariakaloumenou1@gmail.com (M.K.); dtsouk@central.ntua.gr (D.T.); 2Medical School, National and Kapodistrian University of Athens, 75, Mikras Asias Str., Goudi, 11527 Athens, Greece; nlagopati@med.uoa.gr (N.L.); stathise@med.uoa.gr (E.E.)

**Keywords:** breath analysis, volatile organic compounds, sensors, nanomaterials, differential diagnosis

## Abstract

Early-stage disease diagnosis is of particular importance for effective patient identification as well as their treatment. Lack of patient compliance for the existing diagnostic methods, however, limits prompt diagnosis, rendering the development of non-invasive diagnostic tools mandatory. One of the most promising non-invasive diagnostic methods that has also attracted great research interest during the last years is breath analysis; the method detects gas-analytes such as exhaled volatile organic compounds (VOCs) and inorganic gases that are considered to be important biomarkers for various disease-types. The diagnostic ability of gas-pattern detection using analytical techniques and especially sensors has been widely discussed in the literature; however, the incorporation of novel nanomaterials in sensor-development has also proved to enhance sensor performance, for both selective and cross-reactive applications. The aim of the first part of this review is to provide an up-to-date overview of the main categories of sensors studied for disease diagnosis applications via the detection of exhaled gas-analytes and to highlight the role of nanomaterials. The second and most novel part of this review concentrates on the remarkable applicability of breath analysis in differential diagnosis, phenotyping, and the staging of several disease-types, which are currently amongst the most pressing challenges in the field.

## 1. Introduction

Disease diagnosis is conventionally conducted using expensive, time-consuming, invasive techniques, applied by appropriately trained health care professionals. For instance, gastroscopy, laryngoscopy, and coronary angiography are used for gastric cancer (GCa), lung cancer (LC), and myocardial infraction diagnosis, respectively [1]. Other commonly used methods, such as computed tomography [2] or mammography, used for breast cancer (BC) [3], may also be harmful due to radiation exposure. As a result, patient compliance and utilization of such diagnostic methods are remarkably reduced for a significant part of the population. However, disease and especially cancer early-stage diagnosis via effective high-risk population screening, renders treatment easier [4]. For this reason, ameliorated diagnostic methods are imperative.

Metabolomics, one of the ‘-omics’ disciplines that have progressively become a promising diagnostic tool in medical research, offer a comprehensive analysis of the metabolites contained in biological samples by the combination of analytical techniques with bioinformatics [5]. On the other hand, the term volatolomics is referred to the chemical processes that correlate with volatile organic compounds (VOCs) emitted by body fluids [6], such as peripheral blood, urine, and sweat as well as feces, nasal mucous, gaseous skin excretions, and exhaled breath [6,7,8]. Apart from VOCs (e.g., acetone, isoprene, ethane, pentane), inorganic gases (e.g., CO_2_, CO and NO) and non-volatile compounds/exhaled breath condensates (e.g., peroxynitrite, cytokines, and isoprostanes) constitute the human breath [1,9,10]. Decreased sample complexity, the highly developed appropriate analytical techniques, and the ability of direct or continuous breath analysis using gas sensors render exhaled breath as an exceptional source of gas-biomarkers (VOCs predominantly but also inorganic gases) [8,11]. More than 2000 VOCs have been detected in the exhaled breath [12] and appertain to hydrocarbons, alcohols, aldehydes, ketones, esters [12,13], ethers, carboxylic acids, heterocyclic hydrocarbons [12], aromatic compounds, nitriles [12,13], sulfides, and terpenoids [12] and may be endogenous or exogenous.

Exogenously originated VOCs are correlated with the environment and the habits of the person [14]. VOCs related with cleaning fluids, personal care products, plastic-related VOCs [14], blazes, or air pollution due to industrial/transport gas emissions [12] enter human organism through extended inhalation and are excreted via exhaled breath. Smoking [15], food habits and food supplements, drinks, or medication also consist important sources of VOCs [14,15]. Other important confounding factors affecting the profile of exhaled VOCs are age, gender, ethnicity, living place, and lifestyle [15,16]. Consequently, immediate and recent environmental exposure should be taken into consideration during breath analysis [14].

Endogenously created VOCs comprise high vapor pressure (body and room temperature (RT)) (fragments of [14]) byproducts of normal or pathophysiological metabolic pathways [7,14,17], as well as of microbiome metabolism [14]. They are produced either in the airway region or in other parts of human body, representing the metabolism of the whole organism. In the first case, VOCs are released in the exhaled breath in a direct way [17]. In the second case, produced VOCs enter and circulate in the bloodstream, and, during gas exchange in the alveoli or the airways, excretion to the exhaled breath occurs [15,17] via the alveolar pulmonary membrane [9,10]. Depending on blood solubility, VOCs are exchanged in different sites of the respiratory tract. Nonpolar VOCs with poor blood solubility (blood–air partition coefficient (*λ_b:a_*) < 10) are exchanged in the alveoli, in contrast to blood soluble VOCs (*λ_b:a_* > 100) that are exchanged in lung airways. VOCs of intermediate solubility (10 < *λ_b:a_* < 100) undergo pulmonary gas exchange in both sites [12,15]. Oxidative stress, lipid peroxidation, and reactions catalyzed by cytochrome p450 (CYP450), and liver enzymes are the main biochemical processes correlated with endogenous VOCs [15,18,19]. The correlation of oxidative stress and airway inflammation with exhaled VOCs is summarized in Figure 1. Different VOC classes are potentially correlated with different biochemical reactions and exogenous sources.

VOCs of exhaled breath are regarded as normal [12]. However, concentration differences for some exhaled VOCs could potentially be associated with an abnormal condition of the body [12], as the metabolic processes producing the VOCs are altered in a distinctive way by different diseases [1,10]. Disease-related concentration alternations conventionally concern a group of VOCs rather than a single compound [17]. Apart from this, the concentration of a single compound may vary due to more than one pathophysiological processes, thus being non-specific [20]. Consequently, diagnosis of complex, heterogeneous diseases is scarcely achieved by the recognition of one characteristic stand-alone VOC [7,20]. A mixture of exhaled VOCs, called VOC pattern or ‘breathprint’, consists the signature of a specific disease, correlating with the underlying pathophysiology; this pattern should therefore be recognized so as to achieve disease diagnosis [7,17,20].

Thus, the analysis of the exhaled breath holds a great promise for non-invasive early disease diagnosis [1,21]. Several diseases are investigated via breath analysis [7] for potential diagnosis, the main types of which are presented on Table 1. Specifically in the case of infections by specific species or strains of bacteria, the combination of bacteria-derived VOCs [22] and VOCs produced by the host due to immune response to bacterial antigens as well as VOCs formed due to the host response to bacterial products/metabolites (and vice versa) is detected. Differentiation of the origin of those VOCs is not clinically important [23]. Recently, research interest focused on the diagnosis of SARS-CoV-2 viral infection via breath analysis with remarkable results [24,25,26], employing a diagnostic test, i.e., “BreFence Go COVID-19 Breath Test System” developed by Breathonix [27] and “TracieX Breathalyser” developed by Silver Factory Technology, which have been already provisionally approved by the HSA [28,29]. Apart from early diagnosis, screening of high-risk populations and assessment of therapy efficiency can be permitted using breath analysis, due to being an inexpensive [6] and rapid method, characterized by increased patient compliance [30].

For the analysis of exhaled VOCs and disease diagnosis, two different methods can be used; analytical techniques and sensors. Gas chromatography combined with mass spectrometry (GC-MS) comprises the gold-standard method for the analysis of VOCs patterns in exhaled breath [1,17,33]. Both quantitative analysis (characterized by high sensitivity in the ppb to ppt range [1]) and qualitative analysis (providing information concerning the potential metabolic disease pathways [33]) can be achieved [4,17]. For the detection of very low concentrations of VOCs, the pre-concentration of the breath sample is imperative. Pre-concentration techniques commonly combined with GC-MS include thermal desorption [31,33] (using sampling bags/sorbent tubes [33], mainly Tenax tubes [31]), headspace solid-phase microextraction (HS-SPME) [10,31] (using silica fibers, coated with polymeric nanofilm, mainly Carboxen (CAR)/Polydimethylsiloxane (PDMS) [31]) and the needle trap device [31,34] (sorbent polymer—CAR, PDMS, and/or divinylbenzene (DVB)—packed in a needle [31]). Apart from GC-MS, selected ion flow tube-mass spectrometry (SIFT-MS), ion mobility spectrometry (IMS), proton transfer reaction-mass spectrometry (PTR-MS) [17,35], proton reaction transfer time-of-flight mass spectrometry (PRT-TOF-MS) [1], GC coupled with ion mobility spectrometry (GC-IMS), and flame ionization detector (GC-FID) [31] are also common techniques for breath analysis. However, spectrometry and spectroscopic methods exhibit important limitations [15], such as the use of bulky and expensive equipment by appropriately trained personnel, while the analysis is also time-consuming [15,36], providing no real-time results [33]. In addition, pre-concentration methods, required before the analysis, could potentially lead to sample loss/contamination [34,36]. Thus, despite the advantages of those analytical techniques, their use in clinical practice for point-of-care [37] or screening [33] is limited.

During recent years, however, sensors and e-Noses, have exhibited the prospect of becoming strong diagnostic tools via breath analysis and are rising up to the existing clinical challenges [34,35,37]. Gas sensors comprise inexpensive and simple [12,34,37] easy-to-use devices that are small in size and thus portable [12,34]. In addition, short response time and direct acquisition of results, as well as short sensor recovery time [12], render gas sensors attractive for point-of-care and personalized screening, diagnosis, and disease follow-up [34]. Exhaled gas-target analysis using sensors can be achieved by two different approaches (Figure 2). In the first case, a targeted approach is applied, using a selective mechanism [7,14]. The target is recognized by a selective chemical sensor, designed to measure this single compound in a complex mixture, based on lock-key mechanism [7]. Such selective sensors have been developed for NO, NH_3_, acetone, H_2_O_2_ [6,15], H_2_S, and CH_3_SH [15]. On the other hand, the detection of a unique gas-target pattern, rather than a single exhaled compound, is also possible. Semi-selective/ cross-reactive sensors are artificially intelligent nano-arrays [15] mimicking natural sensing systems [38,39] and are also called “electronic noses” (e-Noses), “artificial olfactory systems” (gas analytes), or “electronic tongues” (liquid analytes). The array consists of distinctive sensors that respond to all/large part of the components of a complex mixture [38], at the same time, in a complementary way [39]. Due to their diversity each individual sensor of the array responds differently (yet not chemically selectively) to a given mixture. Statistical pattern-recognition algorithms and classification techniques are used for the establishment of analyte-specific response patterns, combining the responses of the sensor array elements [38]. It is worth noting that even analytical techniques, e.g., GC-MS, are progressively used for the analysis of total patterns of VOCs, instead of targeting a stand-alone biomarker [17].

Multivariate data analysis, a fundamental tool in breath analysis, improves the human perception of experimental data [16]. Response data obtained after sensor array exposure to a complex chemical mixture are processed by multivariate data analysis [40] in order to assess the discriminating ability of the sensor array [16,41], as well as for the elimination of potential confounding variables (i.e., environmental temperature and humidity) [40]. Multivariate data analysis is also useful for breath analysis using analytical techniques, permitting the identification of the most discriminant VOCs between the different groups studied [32]. Numerous multivariate analysis methods are used in e-Nose systems, including canonical discriminate analysis (CDA), partial least squares regression (PLS regression), discriminant function analysis (DFA), and principal component analysis (PCA) (Figure 3) [40]. PCA comprises the most commonly used method in e-Nose systems [40], while DFA is also frequently used.

PCA is an unsupervised learning technique in which the multidimensional data space is reduced to its main components [16]. Linear combinations of original data (i.e., sensor values) capturing the maximum variance between all data points are acquired [16,42], leading to a reduced set of variables [41], called principal components (PCs). The differentiation of the PCs maintaining most of the original data information from the PCs with the minimum effects, which are excluded, is achieved by employing an appropriate algorithm [41]. PCs define new orthogonal axes for the representation of multidimensional data only in two or three dimensions [16,42]. Thus, a visualized statistical analysis is obtained, permitting discrimination of otherwise entangled data [41]. PC1 is characterized by the greatest response variance, while the magnitudes of variance are diminished from PC2 to PC3 and so on [16,42]. DFA is a linear supervised pattern recognition method, also used for multidimensional experimental data reduction [43]. DFA aims to separate known data groups to the best possible extent [44]. The determination of the discrimination classes is conducted prior to the analysis [43]. Input variables are linearly combined to achieve the maximum variance between classes and the minimum variance for each class [43,44]. DFA output is a set of canonical variables (CVs) that fulfill the above two requirements. The first CV is the most powerful discriminating dimension [43].

In order to evaluate sensor array suitability for disease diagnosis and to assess the ability to correlate VOCs with the appropriate disease, machine learning techniques [43] such as support vector machines (SVM), k-nearest neighbor (k-NN), and artificial neural networks (ANN) [40] are also needed. Machine learning techniques and artificial intelligence in general aim at enabling machines to mimic a specific behavior by designing and developing algorithms based on empirical data that are representative of the relation between the observed attributes [45]. The automated “learning” of complex patterns and intelligent decision-making based on current data comprise one of the principal research fields in machine learning [45] and are especially useful in the development of e-Noses for disease diagnosis. ANN is a commonly used machine learning technique, inspired by the human nervous system. The most discriminant sensing features are firstly determined and comprise the input data of the ANN. The input data are connected with the output (i.e., the classification of samples to specific disease) by the ANN, using a set of appropriate functions. The classification from the available inputs is improved by optimizing specific parameters, such as the number of neurons (calculation centers) that are responsible for the system calculations [43].

A number of requirements have to be fulfilled for the use of a sensor in breath analysis and therefore disease diagnosis. Reproducibility [6], high sensitivity, and good resolution [15], are three fundamental parameters. Low limit of detection (LOD) [13,15] (ppb [38]), wide range of response [38], and increased selectivity are also of great importance, in order to detect the exhaled VOCs in the presence of water vapor found in the humidified clinical samples [15]. The stable baseline in the absence of gas-target biomarkers [6], short response, and recovery times are imperative [13,15], while full recovery of the sensor after analyte removal is also essential. Alternatively, disposable sensors that are simple, cost-effective, and therefore suitable for mass-production can be used [15]. Last but not least, operation at RT is important [38].

In the context of this fast-growing field, novel nanomaterials and sensing devices are constantly being developed. The first part of this review aims to provide an up to date overview of the main classes of sensors investigated for the analysis of exhaled gas-analytes (i.e., VOCs, inorganic gases) for disease diagnosis. After highlighting the importance of nanomaterials, examples of applications of either selective or cross-reactive self-developed and commercial sensing devices are discussed. The second and final part of the paper discusses and highlights the capacity of breath analysis towards differential diagnosis, disease staging and phenotyping. There is currently a strong and urgent need for clinically applicable, portable and non-invasive diagnostic tools that can lead to the distinction of different diseases with similar symptoms; the current review paper evaluates the potential of sensing devices of varying technology, as well as analytical techniques, to serve as reliable breath analysis tools for differential diagnosis while it also discusses advantages and weaknesses that need to be addressed.

## 2. The Role of Nanomaterials

More recently, research interest has notably focused on the development of chemical sensors incorporating nanomaterials [6,46,47,48]. Despite their inability for qualitative analysis in complex samples and poor quantitative performance as well as humidity sensitivity and relatively short life, nanomaterial-based gas sensors possess major advantages that render them exceptionally promising [12]. The small dimensions of nanomaterials (typically 1–100 nm) increase their surface-to-volume ratio and the interaction sites [6,15]; those novel interfaces [38,49] lead to high sensor sensitivity as well as small response/recovery times [6,15]. Higher specificity for a desired analyte is also achieved, by sensibly selecting the physical/chemical properties of the nanomaterials [6,15]. Those properties are attributed to the size, shape, and composition of nanomaterials [38] and can therefore be easily modified [6,15]. On the other hand, similar dimensions between nanomaterials and biomolecules render the former attractive for application in medical diagnostic devices [38]. Last but not least, the combination of nanomaterials with different properties (easily accomplished by large-scale manufacturing methods) permits a synergistic sensing ability for the device [6,15], along with simple, portable and energy-efficient operation [6]. As it will be discussed below, different nanomaterials have been used in several gas sensors for gas-analyte detection, as transduction elements [6].

### 2.1. Metallic Nanoparticles (MNPs)

Noble metals (Au, Ag, Pt, Pd) possess exceptional chemical, physical and electronic properties [50]. They exhibit increased conductivity [13], mechanical robustness [50], oxidation resistance and, thus, chemical stability [13,50]. In the form of nanoparticles (NPs), additional novel properties—attributed to the increased surface area and the domination of quantum-mechanical properties [51]—are exhibited; optical (localized surface plasmon resonance (LSPR) phenomenon) [36] and electrical [51] are amongst the more interesting. The mechanical, optical, and electrical properties of MNPs are composition, size, periodicity and inter-particle distance dependent [51]. Notably, chemical environment-dependence of those properties renders MNPs promising for gas-sensing [36,50].

MNPs are extensively investigated in gas sensors for breath analysis applications and are usually combined with other nanomaterials (e.g., carbon nanotubes (CNTs), graphene, metal oxide semiconductors (MOS)), to form more effective sensing materials, with increased sensitivity and selectivity [13,36]. In addition, gas-sensor sensitivity is overall enhanced by the presence of NPs on the surface of other sensing nanomaterials, due to defect formation on which gases can preferably adsorb. Smaller NPs lead to greater surface area, increasing the number of defects and ultimately sensor sensitivity [52]. MNPs can be also functionalized with a variety of organic ligands hence forming thin films with tunable chemical selectivity (molecularly-capped NPs, MCNPs), characterized by controllable inter-particle distance and reproducible production [38].

### 2.2. Metal Oxide Semiconductors (MOS)

MOS are commonly used in different sensor types as sensing materials, for oxidizing or reducing gas detection. Transition MOS (e.g., NiO, Cr_2_O_3_) are more efficient for gas sensing applications than the non-transition MOS (ZnO, SnO_2_), due to more than one preferred oxidation states [7]. Increased affinity of MOS for negatively charged oxygen species (e.g., O_2_^−^, O^−^), in contrast to compound semiconductors (e.g., GaAs) [53], leads to the creation of surface-trapped charge density; in this way electron depletion layers are formed in n-type semiconductors while hole accumulation layers are formed in p-type semiconductors [7,53]. Gas interaction with the oxygen species alternates the surface charge density, resulting in a resistance change [7]. Specifically, oxidation of reducing gases by the adsorbed oxygen species leads to electron transfer towards the semiconductor surface. The adsorption of oxidizing gases, on the other hand, leads to the removal of electrons from the semiconductor surface [53].

N-type MOS, e.g., SnO_2_, ZnO, WO_3_, TiO_2_, MoO_3_, In_2_O_3_, and Fe_2_O_3_, as well as p-type MOS, such as Co_3_O_4_, CuO, NiO, Cr_2_O_3_, and Mn_3_O_4_, have been used in gas sensors for disease diagnosis [53]. MOS sensitivity is affected by numerous parameters, such as morphology, porosity, particle size, film thickness and doping of MOS, decoration with noble metals, as well as operation temperature [54]. Consequently, various MOS based gas-sensors of different structures (villi-like [55], nanotubes (NTs) [56], Hemi tubes [57], hierarchical fibers [58], nano-sheets [59], or NWs [60]), usually combined with noble MNPs [56,57,58,59], 2D-materials [57], or MOS nanoparticles [59,60] have been investigated as diagnostic tools for a variety of diseases. Notably, in the case of semiconducting nanomaterials, CNTs, graphene-based nanomaterials, and MOS have exhibited the highest response towards acetone, thus holding great promise for diabetes diagnosis [53]. However, despite their enticing applications MOS possess important limitations including high temperature operation [13] (150–500 °C) and confinement to single gas detection, due to lack of selectivity towards polar-nonpolar compounds [7].

### 2.3. Carbon Nanotubes (CNTs)

CNTs are investigated as gas sensing materials for their interesting electrical, mechanical, optical and thermal properties [13,61], as well as for their compatibility with other nanomaterials for enhanced performance [7]. CNTs are divided into single wall CNTs (SWCNTs) with a diameter in the range of 1–5 nm [50] and multi-wall CNTs (MWCNTs) with a diameter in the range of 5–100 nm [7] and an interlayer spacing of 3.4 Å [50]. Concerning their electrical properties, which are most commonly exploited, SWCNTs act as metallic conductors or semiconductors (depending on the chiral angle between hexagons and tube axis), while MWCNTs behave as metallic conductors, with current density up to 10^9^ A/cm^2^ [7].

CNTs–analyte interaction may include Van der Waals or donor–acceptor interactions. However, gas adsorption-provoked charge-transfer is the main sensing mechanism since SWCNTs specifically can function as p-type semiconductors [7,13,50]. Oxidizing gas-adsorption, such as NO_2_, decreases sensing-layer resistance, due to electron withdrawing by the gas. In contrast, reducing analyte-adsorption, such as NH_3_, increases resistance [13]. Remarkably, in those cases SWCNTs act as both the sensing element and the transducer [50].

Both categories of CNTs, however, are characterized by lack of chemical selectivity, high H_2_O affinity, low sensitivity for nonpolar compounds [38], and slow recovery [7]. For this reason, functionalization is imperative. Decoration with MNPs for enhanced selectivity is discussed in the literature [38]. More importantly, CNTs are commonly functionalized with analyte specific entities, covalently (esterification, amidation of carboxyl groups added to CNTs) or non-covalently (supramolecularly, via Van der Waals, and π–π interactions) [50]. Modification of CNTs with non-polymeric organic layers [38] (e.g., polycyclic aromatic hydrocarbons [62]) or polymers [50] have been used for the development of effective cross-reactive gas sensors as diagnostic tools via breath analysis.

### 2.4. Nanowires

NWs have also been investigated for gas sensing systems with SiNWs being the most common while MOS-NWs are also used. SiNWs possess interesting optical and electrical properties and compatibility with the technologies currently used in microelectronics [38], acting as n-type or p-type semiconductors and with a maximum operation temperature of 150 °C. Conductivity is modified depending on the nature of the gaseous analyte (oxidizing/reducing) and the type of SiNWs (n-/p-type), while the adsorption of charged oxygen species (O_2_^−^ at 150 °C) determines the NWs conductance properties [63]. VOC polarity is of great importance while the physical adsorption of polar molecules via Van der Waals/electrostatic interactions affects the surface potential [7,63].

Modification of SiNWs properties is feasible [7]; SiNWs doping determines sensor sensitivity by affecting the number of charge carriers of the NWs, similarly to MOS [63]. Chemical functionalization with appropriate molecular ligands [43,64] on the other hand enhances sensor selectivity since non-polar vapor detection by pristine SiNWs is inefficient in contrast to polar VOCs [7]. Molecular modification also serves in the fabrication of cross-reactive arrays for the detection of potential VOC biomarkers [38,43,65,66].

### 2.5. D-Materials

2D-materials (e.g., graphene, MoS_2_, MoSe_2_, WSe_2_, and NbSe_2_) have been investigated for gas sensor development. Their main advantage is low-power sensor operation at RT, while they are characterized by unique electrical properties [13] and large surface-to-volume ratio [13,67]. The latter renders 2D materials excellent candidates for gas sensing applications [67]. Gas analyte adsorption at multiple active sites, on the edge and surface defects of the material, changes its electronic properties. Similarly to CNTs charge-transfer interactions comprise the basis of the gas sensing mechanism [13]. Gas adsorption changes the resistivity of the sensor, depending on the type of gas (reducing/oxidizing) and semiconducting 2D-material (p-/n-type) [67].

Transition metal dichalcogenides and graphene in particular are extensively used in gas sensing systems, facilitating disease diagnosis via breath analysis [7,53,67]. Transition metal dichalcogenides (TMDs) (MX_2_, M: transition metal, X: S, Se, or Te) such as MoS_2_, MoSe_2_, Wse_2_, NbSe_2_, [13] WS_2_, SnS_2_, and SnSe_2_ [68], possess great structural and optical features [69] and tunable semiconducting properties [68] that render them promising materials for e-Noses. TMDs are able to create flexible structures and to operate at low temperatures; however, the increased response/recovery times and the inability to fully recover after analyte exposure are important limitations [68]. To cope with these issues, catalytic MNPs which also improve selectivity can be employed [68,69].

Graphene is characterized by exceptional electrical, thermal and mechanical properties, with low resistivity of 10^−6^ Ω∙cm at RT and large surface-area of 2630 m^2^/g (higher than CNTs, 1000 m^2^/g [53]) being the most promising 2D material for gas sensing. Graphene can also behave as a p-type semiconductor while chemical functionalization can be easily achieved [7,68]. Remarkably, graphene has been used for selective rather than cross-reactive applications. Graphene derivatives have also been used for gas sensing applications. Graphene oxide (GO) is characterized by a lower cost of production, however its application in electronic devices is hindered by reduced conductivity. Reduced graphene oxide (RGO) on the other hand exhibits tunable conductivity along with greater gas responses than graphene since the presence of oxygen functional groups enhances gas adsorption [67,68]. It is noteworthy that the use of RGO-based gas sensors needs further investigation as the underlying gas-oxygen groups interactions are yet to be explained [53].

2D materials are usually combined with various nanomaterials, for enhanced sensing [7]. The poor sensing performance of pristine 2D materials is primarily attributed to the weak interactions between the adsorbed molecules and the sensing layer [13]. Graphene and RGO combined with MOS [57,70] and MNPs [59] may serve in the development of gas sensing diagnostic tools.

### 2.6. Hybrid Materials

The combination of distinctive nanomaterials so as to form hybrid materials is extensively investigated for disease diagnosis via gas-sensing, as previously discussed. The combination of different nanomaterials improves sensing selectivity and sensitivity [7,13,38] due to their synergic action, rendering hybrid materials particularly promising [49]. Combinations of CNTs with MNPs, 2D materials with MOS or MNPs as well as MNPs [41,71], MOS NPs, or CNTs with polymers are only some of the reported combinations [7]. Particularly polymers are commonly used for achieving chemical selectivity. Concerning (semi-)conducting polymers (e.g., polyaniline, polypyrrole, polythiophene), conductivity changes can be attributed to electrically-active analyte adsorption, due to redox interactions with either the backbone or the incorporated particles in the case of composites [50]. Non-conducting polymers are also used as sensing films, incorporating conductive NPs such as CNTs [50] or carbon black particles [72] for semi-selective analyte absorption; this leads to mass/conductivity changes of the polymeric nano-composites. Finally, molecularly imprinted polymers (MIPs) are a new class of sensing films in which artificial analyte-specific cavities have been created, for specific molecular recognition [73,74].

## 3. Types of Nanomaterial-Based Sensors in Breath Analysis

Different sensor types have exhibited promising diagnostic ability via breath analysis [7]. In the following part nanomaterial-based gas sensors are categorized depending on the transduction method. Examples of both selective and cross-reactive applications are presented, in an attempt to provide an up-to-date overview.

### 3.1. Chemiresistors

Chemiresistors are one of the most promising types of gas-sensors. They are characterized by simple configuration and working principle [13], increased reliability, decreased size and weight, while they are suitable for automatic packaging in wafer level thus permitting the mass-production of portable, on-chip sensor arrays [51]. Two pairs of electrodes are connected with an overlying sensing layer that is either semiconducting or metallic. Interdigitated electrodes (IDEs) are commonly used for enhanced sensing response [13]. Sensor-resistance changes upon gas exposure, while constant current/potential is applied between the two electrodes [13,51]. Quantitative analysis of the analytes is possible by measuring the change of resistance/current [13]. A schematic representation of typical chemiresistors is presented in Figure 4.

#### 3.1.1. Selective Applications

Concerning single analyte detection, nanomaterial-based sensing films such as MOS, CNTs and hybrid materials are commonly used. MOS-based chemiresistors, in particular, (e.g., WO_3_ [55], SnO_2_, or Cu_2_O [36]) are common in the detection of oxidizing/reducing gases [75]. Moon et al. [55] used a chemiresistive porous thin film composed of villi-like WO_3_ structures and achieved selective and sensitive detection of the asthma biomarker NO in presence of ethanol, acetone, NH_3_ and CO (80% relative humidity (RH)) at 150–250 °C, with a LOD of 88 ppt which is far lower than exhaled NO. Selective arrays of MOS sensors have been also used; highly porous Pt, Si, Pd, or Ti doped SnO_2_ NPs have selectively detected low levels of formaldehyde (3 ppb) in 90% RH, as well as in synthetic breath samples containing acetone, ethanol and NH_3_, aiming at LC diagnosis [76]. Concerning CNT-based chemiresistors, NO_2_ selective detection has been reported, using a random or an aligned network of SWCNT, with the former achieving a lower LOD in the range of ppt [77]. Ethanol detection by vertically-aligned CNTs has also been achieved [78].

Among hybrid materials, MNPs-functionalized nanomaterials are commonly investigated. Highly porous thin-wall SnO_2_ fibers of hierarchical structure composed of Pt NPs-decorated SnO_2_ NTs, have been investigated for acetone and toluene detection as diabetes and LC biomarkers respectively; increased sensor responses for both analytes were permitted, due to MNPs-functionalization [58]. More recently, Pt-decorated Zn_2_SnO_4_ hollow octahedra were used for enhanced selective detection of acetone, in ppb levels, adequate for breath analysis applications and diabetes diagnosis [79]. In another application, Au NPs-decorated SWCNTs were used for selective detection of H_2_S (halitosis biomarker) in the presence of NH_3_ and NO, with a LOD of 3 ppb [80]. Au NPs functionalization permitted increased sensitivity for the desired analyte [80].

Other types of combinations of nanomaterials, including MOS, CNTs, or 2D materials, have been also studied. The respective halitosis and diabetes biomarkers H_2_S and acetone for example, have been detected using rGO nanosheets-functionalized SnO_2_ NFs with increased selectivity and sensitivity [81] as well as using GO-WO_3_ and thin-layer graphite-WO_3_ Hemi tubes in the presence of CO, NH_3_, NO, ethanol, and pentane; high responsiveness and sensitivity and a lower LOD for H_2_S were achieved in the latter study [57]. The combination of graphene derivatives with conducting polymers was also reported; porous and mechanically improved films composed of GO and polypyrene for instance, have been used for rapid toluene detection [82]. In the field of CNTs-based hybrid materials, SWCNTs were recently combined with porphyrin-based NFs to form a nano-composite of 1D nano-architecture that permitted the selective detection of H_2_O_2_, in the simultaneous presence of various gases and VOCs. The enhanced sensitivity, reproducibility, and responsiveness, rendered the sensing film promising for cancer, traumatic brain injury, chronic obstructive pulmonary disease (COPD), and asthma diagnosis [83]. The combination of CNTs with conducting polymers has been reported as well, especially for selective detection of ammonia. Such sensors employ carboxylated-MWCNTs-polypyrrole [84] or polyaniline-MWCNTs nano-composites featuring increased sensitivity in the presence of H_2_S, NO_2_ acetone, ethanol, and isoprene [85].

#### 3.1.2. Cross-Reactive Applications

Apart from selective applications, chemiresistors have also been extensively investigated for the development of sensor arrays and cross-reactive devices. MOS have been used as components of such sensing systems. Arrays of commercial MOS sensors (TGS), have been used by Binson et al. [86,87] and Marzorati et al. [88] for the differentiation of LC [86,87,88] and COPD patients from HC, with high accuracies (Table 2). Interestingly, LC diagnosis using an array of TGS sensors has been reported in the literature; notably, 85 real-world breath samples of LC patients have been used, aiming not only at the discrimination between LC and HC but also at before- and after-surgery patients [89]. The aforementioned system could potentially comprise a promising diagnostic and prognostic tool after LC resection surgery, since it was shown that HC and operated patients’ breath-prints converged over time Table 2 [89]. It is noteworthy that commercial MOS-based chemiresistive e-Noses have been developed and studied for breath analysis and diagnosis. “Common Invent e-Nose”, for instance, has been used for asthma phenotyping and exacerbation prediction [90], while “SpiroNose” [91] and “^AeoNose®^” [92] have been used for asthma, COPD and LC patients and COPD, LC patients and HC differentiation, respectively. “DiagNose”, composed of 12 different doped MOS-sensors, comprises another promising example since it managed to differentiate 10 individuals through their breath samples [93]. Self-developed MOS-chemiresistors, in the form of pellets, based on a mixture of SnO_2_ and ZnO nano-powders (9:1) have been effectively used for asthma diagnosis; however, a small number of subjects was included [94]. The combination of various materials for sensor-array development was also reported; an on-chip array of parallel ZnO, Pd, and polypyrrole single NWs for instance has been used for BC biomarkers, permitting increased responsiveness and sensitivity as well as total discrimination of the volatile biomarkers (Table 2) [95].

However, hybrid materials comprise the most extensively used materials for chemiresistive sensor arrays. Noble metal-functionalized MOS have been broadly studied. Halitosis diagnosis using an array of pristine and Pt NPs-decorated WO_3_ macroporous NFs comprises a representative example (Table 2); increased responses towards H_2_S were achieved by the 0.042 wt% Pt NPs-WO_3_ NFs [96]. PCA successfully discriminated H_2_S, acetone, toluene, and methyl mercaptan vapors, while real-breath samples of healthy controls (HC) and simulated halitosis breath were effectively classified [96]. Similarly, an array of pristine Au NPs (1/3 nm) decorated with vertically ordered hematite NTs, has been studied for acetone detection; the desired analyte led to far greater responses, while the studied vapors (acetone, ethanol, CO, H_2_, NH_3_, toluene, and benzene) were differentiated using PCA [97]. The theoretical acetone LOD of 304 ppt renders Au NPs-Fe_2_O_3_ NTs-based sensor particularly promising for diabetes diagnosis. Cross-sensitivity towards NO and toluene using pristine WO_3_ NTs (with increased selectivity for NO) and catalyst (Pt and Pd) decorated WO_3_ NTs (more selective for toluene) has been also achieved, holding great promise for asthma and LC diagnosis via breath analysis [56]. The development of halitosis and diabetes diagnostic sensors has been attempted by the same group using cross-sensitive, pristine, thin-walled, and Pt NPs-decorated WO_3_ Hemi tubes, with increased sensitivity towards H_2_S and acetone respectively [98]. More recently, a 2 × 4 array of sensors based on pristine Pt, Pd, or Au NPs decorated In_2_O_3_ and WO_3_ nano-rods (NRs), was investigated for the detection of acetone, NO, and H_2_S as diabetes, asthma, and halitosis biomarkers. This achieved low LOD and visual discrimination between the three analytes using polar plots (Figure 5); however, further statistical analysis is required [99]. The same group has also developed a 3 × 3 sensor array based on WO_3_, SnO_2_, or In_2_O_3_-based thin films, Au NPs-decorated thin films and Au NP-decorated villi-like structures; the array efficiently discriminated ppt levels of H_2_S and ppb levels of NH_3_ and NO in 80% of RH, being potentially applicable to halitosis, renal disorder, and asthma diagnosis, respectively [100]. The notable sensor sensitivity was attributed to the increased surface-to-volume ratio created by the Au NPs and the highly porous villi-like structure [100].

Apart from MNPs-decorated MOS, other promising types of hybrid materials have been studied; MOS for instance have been combined with graphene derivatives. An e-Nose based on graphene-doped TiO_2_, NFs, and nano-ribbons was recently developed for ethanol, acetone, CO and NO detection, common biomarkers for *Staphylococcus aureus* infection, diabetes, asthma, COPD, and cystic fibrosis (CF), respectively [101]. Breath-simulating samples of different VOCs-concentrations (respective to HC and patients of a selected disease) have also been studied, leading to effective discrimination of VOC concentrations. In another attempt for the development of an e-Nose, molecularly functionalized rGO layers with different amine-ligands, have been used for the discrimination of exhaled cancer biomarkers ethanol, 2-ethylhexanol, nonanal, and ethyl benzene [102]. The molecular ligands, serving as the organic sensing film, alter the adsorption capacity and conductivity of the rGO. Concerning CNTs-based cross-reactive sensor arrays, calixarene functionalization has permitted the detection of aromatic compounds (toluene, benzene, ethyl benzene, and xylenes) with high sensitivity [103]. Decoration with Au (1.5 nm), Pd (0.2 nm), and Cr (1.0 nm) has been used along with a pristine sensor element to effectively distinguish NH_3_, ethanol, CO, and CO_2_ using PCA [104]. Remarkably, in a more recent study, an array of nylon fibers wrapped with SWCNTs, MWCNTs, and ZnO QDs-SWCNTs has been developed as a flexible wearable sensor to be incorporated in face masks. Detection and effective discrimination of NH_3_, ethanol, and formaldehyde [105], and common disease biomarkers (ethanol [101], formaldehyde [106], NH_3_ [7]) was achieved [7]. SWCNTs-functionalization with semiconducting organic polymers, monomers or oligomers is also reported, especially for real-world sample experiments, discriminating successfully COPD patients from HC as well as selected analytes; however, larger clinical trials remain to be conducted (Table 2) [107].

The most common types of hybrid sensing layers are those composed of a conductive inorganic material, surrounded by an organic functional film where the gaseous analytes are adsorbed therefore changing the conductivity of the device [15]. CNTs, MNPs (e.g., Au) [15], or carbon black [15,108] can be used as the conductive part while non-conducting molecular ligands (MCNPs) [51,108,109] or polymers [110] as the organic one. Such sensors offer the prospect of cross-reactive sensor-array development since the organic part is selected based on the chemical and physical properties of the VOCs [16]. Upon sensor exposure to gaseous biomarkers, the latter diffuse into the organic layer interacting with its functional groups [15]. Those interactions cause swelling of the organic matrix [15], posing a stress to the underlying NPs layer [41], thus increasing the inter-particle distance of the conducting NPs and the measured resistance [6]. In addition, the relative dielectric constants of the VOCs and the organic layer, all of which affect the permittivity of the organic matrix, change the measured resistance [6].

Molecularly-functionalized conducting nanomaterials have been extensively investigated for the cross-reactive detection of VOCs by Haick et al. Molecularly-capped Au NPs and molecularly-coated random SWCNTs networks have been successfully developed for exhaled VOCs-biomarkers detection using real-world samples, usually in combination with chromatographic analysis of breath samples. Irritable bowel syndrome disease (IBS) [111], ovarian [44], colon, lung, gastric [16,112], breast and prostate cancer [16,42], chronic kidney diseases (CKD) [113], multiple sclerosis [62], Alzheimer’s and Parkinson’s diseases [15,114] have been effectively diagnosed with such chemiresistors. In the case of multiple sclerosis e.g., PAH-coated SWCNTs sensor-arrays have exhibited sensitivity, specificity and accuracy percentages comparable to those of invasive and expensive techniques such as MRI and cerebrospinal fluid electrophoresis [62]. VOCs targeted for the diseases mentioned above are presented on Table 2. In a remarkable application of a MCNPs/SWCNTs-based chemiresistor array (20 sensors), 17 different diseases (Table 2) were successfully discriminated, with 86% accuracy, based on the detection of a pattern of only 13 VOC whose concentration differed significantly between HC and/or different diseases [115]. More recently, a MCNPs-based chemiresistor was developed for COVID-19 detection, with remarkable diagnostic accuracy over healthy and non-COVID infected subjects [24]. Despite the fact that targeted VOCs were not listed in this study, the most notable VOCs for COVID-19 are probably methylpent-2-enal, 2,4-octadiene, 1-chloroheptane, and nonanal (10–250 ppb) [25]. Interestingly, a tailor-made nanoscale artificial nose (NA-NOSE), based on this type of sensing films, has been developed by Haick et al. [116]. NA-NOSE is composed of seven cross-reactive MCNPs-layers with different ligands, from which 6 are composed of spherical Au NPs and 1 of cubic Pt NPs. NA-NOSE seems to be promising for breath analysis, as it has been effectively used, e.g., for BC diagnosis [116]. Molecularly-modified Au, Pt, and Cu NPs have been used for the development of a 6-sensors array, to effectively diagnose the infectious disease human cutaneous *Leishmaniasis* with a 98.2% accuracy, after identifying 9 potential biomarkers using GC/quadruple-TOF [109]. A representative example of MCNPs-based chemiresistors is presented in Figure 6.

The size, composition, inter-particle distance, and periodicity of the NPs as well as the aggregate thermal stability of such sensors are essential yet easily controlled parameters [51]. However, one major drawback is humidity cross-sensitivity which is a major component of exhaled breath. Thus, sensor reliability and reproducibility for real samples analysis is of major concern. Humidity compensation is proposed as an effective solution, enhancing the diagnostic ability of the sensor [42].

Remarkably, hybrid composites based on polymers seem particularly promising as well. One of the most commonly studied commercial e-Noses i.e., Cyranose 320, consists of 32 chemiresistors based on composites of carbon black conducting particles and different polymeric films [117,118]. Cyranose 320 has been effectively used amongst others, for the discrimination of asthma [119], COPD [120], BC [121], or pneumonia [122] patients from HC. Among self-developed sensor arrays, CNTs combined with 5 different polymers have permitted the effective differentiation of 9 VOCs as LC potential biomarkers using PCA analysis (Table 2) [110]. Notably, the combinations of MNPs with polymers, apart from molecular organic ligands, comprise a promising type of coating materials for chemical cross-selectivity to be achieved. Chemically unmodified MNPs (Pt [48,123] or Au [71] of mean diameter 4–5 nm [41,48,123]), coated with different polymeric films and with affinity for different compounds, have been investigated along with pattern recognition methods for pesticide detection [41] and more recently pesticide discrimination (chlorpyrifos, bupirimate, and humidity) [124]. The sensing film is deposited on Au IDEs [41] while the device can be fabricated on rigid (e.g., oxidized Si wafer) [41,48] or flexible (e.g., polyimide) [123] substrate as can be seen in Figure 7. Such sensors could be particularly promising for disease diagnosis applications, as semi-selective sensors of exhaled VOCs.

**Table 2 sensors-22-01238-t002:** Sensor arrays for exhaled VOCs detection as biomarkers of several diseases, in real-world or synthetic samples, using conventional materials and/or nanomaterials.

Sensing Element	Disease	Targeted VOCs	LOD	Subjects	Classifier	Results	T	Ref.
**In Vivo Studies—Real-World Samples**								
**Chemiresistor—arrays**								
**Molecularly capped AuNPs—14 different ligands**	Lung cancer	1-Methyl-4-(1-methyl ethyl) benzene, Toluene, 3,3-Dimethyl pentane, 2,3,4-Trimethyl hexane, Dodecane, 1,1′-1-Butenylidene)bis benzene	NA	30 LC, 26 CC, 22 BC, 18 PC, 22 HC	PCA	Good discrimination of cancer types from HC, but not between them. No VOC overlap in abundance between HC and cancer patients.	RT	[16]
	Colorectal cancer	1,1′-(1-Butenylidene)bis benzene, 1,3-Dimethyl benzene, 1-Iodo nonane, (1,1-Dimethylethyl)thio acetic acid, 4-(4-Propylcyclohexyl)-4′-cyano1,1′-biphenyl-4-yl ester benzoic acid, 2-Amino-5-isopropyl-8-methyl-1-azulene carbonitrile						
	Breast cancer	3,3-Dimethyl pentane, 2-Amino-5-isopropyl-8-methyl-1-azulene carbonitrile, 5-(2-Methylpropyl)nonane, 2,3,4-Trimethyl decane, 6-Ethyl-3-octyl ester 2-trifluoromethyl benzoic acid						
	Prostate cancer	Toluene, 2-Amino-5-isopropyl-8-methyl-1-azulene carbonitrile, 2,2-Dimethyl decane, p-Xylene						
**Molecularly capped AuNPs—7 different ligands**	Prostate cancer	Toluene, 2-Amino-5-isopropyl-8-methyl-1-azulene carbonitrile, 2,2-Dimethyl decane, p-Xylene	NA	9 PC, 10 HC	DFA	100% specificity, 100% sensitivity	RT	[42]
	Breast cancer	3,3-Dimethyl pentane, 2-Amino-5-isopropyl-8-methyl-1-azulene carbonitrile, 5-(2-Methylpropyl)nonane, 2,3,4-Trimethyl decane, 6-Ethyl-3-octyl ester 2-trifluoromethyl benzoic acid		10 BC, 11 HC		100% sensitivity, 95% specificity		
**Molecularly capped AuNPs—3 different ligands**	Chronic kidney disease	*healthy* vs. *stage 2:* Isoprene, Acetone, Styrene, Toluene, 2-Butatone, 2,2,6-Trimethyl octane, 2,4-Dimethyl heptane *Stage 2* vs. *3:* Isoprene, Acetone, 2,2,6-Trimethyl octane, 2-Butatone, 2,4-Dimethyl heptane *Stage 3* vs. *4:* Acetone, Ethylene Glycol, Acetoin	1–5 ppb	42 CKD, 20 HC	SVM	79% accuracy early-stage CKD vs. HC 85% accuracy CKD stage 4 vs. stage 5	RT	[113]
**Molecularly capped AuNPs—5 different ligands**	Ovarian cancer	Styrene, Nonanal, 2-Ethylhexanol, 3-Heptanone, Decanal, Hexadecane	ppb level	17 OV, 26 HC	DFA	82% accuracy	RT	[44]
**Molecularly capped AuNPs—8 different ligands**	COVID-19	NA	NA	49 COVID-19, 33 non-COVID symptomatic, 58 HC	DFA	76% accuracy COVID-19 vs. HC 95% accuracy COVID-19 vs. non-COVID symptomatic	RT	[24]
**PAH-coated random SWCNTs network—4 different PAHs**	Multiple sclerosis	Hexanal, 5-Methyl-undecane	NA	37 MS, 18 HC	DFA	80.4% accuracy	RT	[62]
**Molecularly caped AuNPs/CDs-coated random SWCNTs network—20 different sensing films**	Alzheimer’s and Parkinson’s disease	24 VOCs	1–5 ppb	15 AD, 30 PD, 12 HC	DFA	85% accuracy AD vs. HC 78% accuracy PD vs. HC 84% accuracy AD vs. PD	RT	[114]
**Molecularly caped AuNPs/PAH-coated random SWCNTs network—20 different sensing films**	17 diseases (LC, CC, HNC, OC, BLC, PC, KC, GC, CD, UC, IBS, IPD, MS, PDISM, PH, PET, CKD)	2-Ethylhexanol, 3-Methylhexane, 5-Ethyl-3-methyloctane, Acetone, Ethanol, Ethyl acetate, Ethyl benzene, Isononane, Isoprene, Nonanal, Styrene, Toluene, Undecane	10 ppb	813 any of 17 diseases, 591 HC	DFA, HCA	86% average accuracy	RT	[115]
**Ligand capped Au, Pt, and CuNPs—6 different sensing films**	Human cutaneous leishmaniasis	2,2,4-trimethyl pentane, 4-methyl-2-ethyl-1-pentanol, methyl vinyl ketone, nonane, 2,3,5-trimethyl hexane, hydroxy-2,4,6-trimethyl-5-(3-methyl-2 butenyl)cyclohexyl) methyl acetate, 3-ethyl-3-methyl heptane, octane, 2-methyl-6-methylene-octa-1,7-dien-3-ol	NA	28 HCL, 32 HC	PCA, DFA	98.2% accuracy, 96.4% sensitivity, 100% specificity	RT	[109]
**pristine, COOH-, Hex-4T-Hex/DNA/oligomers, PTCDA/TAPC/TCTA monomers or PANI-functionalized SWCNTs**	COPD	NH_3_, NO_2_, H_2_S, benzene, 2-propanol, acetone, ethanol, sodium hypochlorite, water	sub-ppb	12 COPD, 9 HC	PCA	Acetone, ethanol and 2-propanol selective PANI-, TAPC- and COOH-CNTs, respectively. NO_2_ relevant driver of real-samples classification. Larger clinical trials needed.	RT	[107]
**Pristine WO_3_, 0.008 wt %** **and 0.042 wt % Pt-WO_3_ macroporous NFs**	Halitosis	H_2_S and Methyl mercaptan (in presence of Toluene and Acetone)	sub-ppm	4 simulated halitosis breath samples (1 ppm), 4 HC	PCA	Successful classification	350 °C	[96]
**7 different commercial MOS**	Lung cancer	Ethyl benzene, 4-Methyl octane, Undecane, 2,3,4-trimethyl hexane	Down to a few ppb	37 NSCLC (81.1% I, II), 48 HC	PCA	75% accuracy Promising prognostic tool after LC resection surgery	300 °C	[89]
**5 different commercial MOS**	Lung cancer, COPD	NA	NA	32 LC, 38 COPD, 72 HC	PCA, SVM, k-nearest neighbors	LC vs. HC: 91.3% accuracy, 84.4% sensitivity and 94.4% specificity COPD vs. HC: 90.9% accuracy, 81.6% sensitivity and 95.8% specificity	NA	[87]
**Field Effect Transistor (FET)—arrays**								
**Molecularly modified SiNWs**	Gastric cancer	2-Propenenitril, Furfural, 6-Methyl-5-heptene-2-one	Down to a few ppb	30 GC, 77 HC	DFA	>85% accuracy	RT	[125]
**Molecularly modified SiNWs**	Gastric cancer	2-Propenenitril, Furfural, 6-Methyl-5-heptene-2-one	Down to a few ppb	149 LC, 40 GC, 56 Asthma/COPD, 129 HC	DFA, ANN	>80% accuracy	RT	[43]
	Lung cancer	Heptane, Decane, 2-Methyl pentane, 2-Ethyl-1-hexanol, Propanal, Pentanal, Acetone						
	Asthma/COPD	Pentane						
**Electrochemical sensor**								
**Commercial NO, CO sensors, carbon electrode with linear-aldehyde selective porous poly tetrafluoroethylene membrane**	Diabetes	NO, CO, Formaldehyde, Acrolein, Propanal, Crotonaldehyde, Butanal, Pentanal, Hexanal, Heptanal, Octanal, Nonanal, Decanal, Acetaldehyde	Low ppb	15 diabetic, 14 HC		LC vs. HC, diabetic vs. HC Cross-sensitivity for aldehyde sensor: Moderate for high level of ethanol and isopropanol/Weak for H_2_S, NO, methanol, 3-heptanone/None for NO_2_, propofol, isoprene, or acetone	RT	[126]
	Lung cancer			3 LC, 3 smokers, 3 HC				
**Optical—Colorimetric sensor arrays**								
**24 chemically reactive colorants**	Lung cancer	NA	Low ppm	92 LC, 137 HC	LPM	Accuracy 81.1% LC vs. HC, 82.5–89% one histology vs. HC, 86.4% ADC vs. SCC	RT	[127]
**Optical sensors**								
**PMTFP-coated optical fiber**	Vit. E deficiency	Ethane	pmol/L	20 HC	NA	NA	RT	[128]
	Liver diseases, Schizophrenia, Breast cancer, Rheumatoid Arthritis	Pentane						
	Lung cancer	Heptane, Octane, Decane, Benzene, Toluene, Styrene						
**Piezoelectric (SAW) sensor arrays**								
**GC-column/Polyisobutylene-coated SAW, non-coated SAW sensors**	Lung cancer	Styrene, Decane, Isoprene, Benzene, Undecane, 1-Hexene, Hexanal, Propyl benzene, Heptanal, 1,2,4-Trimethyl benzene, Methyl cyclopentane	500 ppb	20 LC, 15 HC, 7 chronic bronchitis	ANN	80% sensitivity and specificity	RT	[6,129]
**Piezoelectric (QCM) sensor arrays**								
**7 different** **metalloporphyrins**	COPD	NA	NA	5 COPD per GOLD stage (20), 5 HC	PLS-DA	Fair repeatability of measurements within HC and hypoxemic COPD patients (stage 4) Potential COPD severity assessment	RT	[130]
**8 different** **metalloporphyrins**	Asthma	NA	NA	27 asthma, 24 HC	PCA, FNN	87.5% accuracy	RT	[131]
**8 different** **metalloporphyrins**	Lung cancer	NA	NA	20 LC, 10 HC	PLS-DA	85% accuracy LC vs. HC 75% accuracy ADC vs. SCC	RT	[132]
**8 different** **metalloporphyrins**	Lung cancer	NA	NA	70 LC, 76 HC	PLS-DA	81% sensitivity, 100% specificity	RT	[133]
**8 different** **metalloporphyrins**	Tuberculosis	NA	NA	51 TB (31/51 +HIV), 20 HC	PCA, k-nearest neighbors	94.1% sensitivity, 90% specificity	RT	[134]
**7 different** **metalloporphyrins**	Halitosis	H_2_S, Butyric acid, Valeric acid	10–15 ppb	Oral malodor subjects, HC	PCA	PC1 78% of data variance	50 °C	[135]
**8 different anthocyanins**	Asthma	NA	NA	15 asthma, 27 HC	Factor Analysis	75% of total variance Repeatability similar to spirometry and eNO	RT	[136]
**In vitro studies—Cell lines/Synthetic samples**								
**Chemiresistor arrays**								
**CNT-conductive polymer nanocomposites—5 different polymers**	Lung cancer	Isopropanol, Tetrahydrofuran, Dichloromethane, Toluene, n-Heptane, Cyclohexane, Methanol, Ethanol, Water		NA	PCA	High sensitivity and selectivity for all the analytes, PC1-PC3 98% of total variance, except the two alkanes	RT	[110]
**Pristine rGO and rGO functionalized with 8 different amine ligands—9 elements**	Cancer	Ethanol, 2-Ethylhexanol, Ethyl benzene, Nonanal	25 ppm	NA	PCA	Successful discrimination of VOCs The LOD and the effect of humidity have to be decreased	RT	[102]
**Pristine Pd, ZnO and polypyrrole NWs**	Breast cancer	Heptanal	8.98 ppm	NA	PCA	73.2% PC1 variance High sensitivity and specificity	RT	[95]
		Acetophenone	798 ppb					
		2-Propanol	129.5 ppm					
		Isopropyl myristate	134 ppm					
**Pristine In_2_O_3_ and WO_3_ NRs, Au, Pt, or Pd NPs-decorated In_2_O_3_ and WO_3_ NRs—8 elements**	Diabetes	Acetone	1.48 ppb	NA	Polar plot	Effective visual discrimination between the gases. Future PCA, DFA, HCA analysis.	150–300 °C	[99]
	Asthma	NO_2_	1.9 ppt					
	Halitosis	H_2_S	2.47 ppb					
**WO_3_ NTs** **Pt NPs—WO_3_ NTs,** **Pd NPs—WO_3_ NTs**	Asthma	NO	50 ppb	NA	NA	NA	350 °C	[56]
	Lung cancer	Toluene	100 ppb				400 °C	
**Pristine, 0.1% wt GO- and 0.1 wt % thin layered graphite WO_3_ Hemi tubes**	Diabetes	Acetone	1 ppm	NA	NA	NA	350 °C	[57]
	Halitosis	H_2_S						
**Electrochemical sensor**								
**MWCNTs/Au-Ag NPs/GCE**	Gastric cancer	3-Octanone	0.3 ppb	MGC-803 GC and GES-1 gastric mucosa cell lines	NA	Easy cell line discrimination, high sensitivity, good VOCs selectivity in presence of CO_2_, acetone and ethanol	RT	[137]
		Butanone	0.5 ppb					
**SiNWs-rGO**	Infectious diseases	Cyclohexane, Formaldehyde in presence of Methanol, Ethanol, Acetonitrile, Acetaldehyde and humidity	1 ppm	NA		Novel electrode platform with increased sensitivity, selectivity and repeatability		[106]
**Piezoelectric (SAW) sensor arrays**								
**SH-Calix4arene,** **AuNRs, AgNCs,** **Calix4arene-AuNRs, Calix4arene-AgNCs**	Lung cancer	Chloroform, Toluene, Isoprene, Acetone, n-Hexane, Ethanol	1.52–12.34 ppm for CHCl_3_ 1.54–2.64 ppm for toluene	NA	NA	Sensitivity ↑ for all VOCs Chloroform, toluene: 6–8 times higher sensitivity than individual responses Selectivity ↑: modified AuNRs for CHCl_3_, modified AgNCs for Toluene	RT	[49]
**Pristine or AuNPs-functionalized zeolitic-imidazole-framework nanocrystals (ZIF-8, ZIF-67)**	Diabetes	Acetone, Ammonia, Ethanol	acetone 1.1–3.6 ppm, ethanol 0.5–3 ppm NH_3_ 1.6–3.2 ppm	NA	PCA	Effective discrimination of diabetes biomarkers	RT	[138]
**Piezoelectric (QCM) sensor arrays**								
**TiO_2_-MWCNTS and Cobalt (II) phthalocyanine-silica on Au layers**	Diabetes	Acetone	4.33 ppm	NA	NA	High sensitivity	RT	[139]
	Asthma	NO	5.75 ppb					
**Optical—Colorimetric arrays**								
**36 chemically responsive dyes (porphyrin derivatives, NaFluo)**	Lung cancer	p-Xylene, Styrene, Isoprene, Hexanal	50 ppb	NA	HCA, PCA, BPNN	100% accuracy of kind and concentration discrimination, promising for real-sample experiments	RT	[140]
**AuNRs-modified metalloporphyrins and pH responsive dyes—36 spots**	Lung cancer	Decane, Undecane, Hexanal, Heptanal, 1,2,4-Trimethylbenzene, Benzene	<1 ppm	NA	PCA, HCA	64.2% accuracy of structurally similar VOCs, 93% photoprotection of metalloporphyrins, ↑ repeatability and long-term stability	RT	[141]

AD: Alzheimer’s Disease, ADC: Adenocarcinoma, ANN: Artificial Neural Network, BC: Breast cancer, BLC: Bladder cancer, BPNN: Back-Propagation Neural Network, BUN: Blood urea nitrogen, CC: Colorectal cancer, CD: Crohn’s Disease, CDs: Cyclodextrin derivatives, CKD: Chronic Kidney Disease, COPD: Chronic Obstructive Pulmonary Disease, DFA: Discriminant Function Analysis, FNN: Feet-forward Neural Network, GC: Gastric cancer, GCE: Glass Carbon Electrode, GOLD: Global Initiative for Obstructive Lung Disease, HC: Healthy control, HCA: Hierarchical Cluster Analysis, HNC: Head and Neck cancer, IBS: Irritable bowel syndrome, IPD: Idiopathic Parkinson’s disease, KC: Kidney cancer, LC: Lung cancer, LOD: Limit of Detection, LPM: Logistic prediction model, MLP: Multi-layer Perceptron, MS: Multiple Sclerosis, MWCNTs: Multi-wall carbon nanotubes, NA: Not Applicable, NaFluo: sodium fluorescein, NCs: Nanocubes, NFs: Nanofibers, NPs: Nanoparticles, NRs: Nanorods, NSCLC: Non-small Cell Lung Carcinoma, OC: Ovarian cancer, PAH: Polycyclic aromatic hydrocarbons, PC: Prostate cancer, PCA: Principal Component Analysis, PD: Parkinson’s Disease, PDISM: Atypical Parkinsonism, PET: Pre-eclampsia toxemia, PH: Pulmonary Hypertension, PLS-DA: Partial Least Square Discriminant Analysis, PMTFP: Polymethyl(3,3,3-trifluoropropyl)siloxane, RT: Room temperature, SCC: Squamous cell carcinoma, SiNWs: Silicon nanowires, SVM: Support Vector Machine, T: Temperature, TB: Tuberculosis, TFB: Poly(9,9-dioctylfluorenyl-2,7-diyl)-co-(4,4′-(N-(4-s-butylphenyl)diphenylamine), UC: Ulcerative Colitis.

### 3.2. Field-Effect Transistors (FET)

Field-effect transistors are voltage-controlled [6] devices consisting of two electrodes (the source and the drain electrode), a semiconducting channel, an insulating gate, and a conducting gate electrode [13]. The current flows between the source terminal and the drain terminal through the semiconducting channel by applying a source-drain potential. Voltage applied between the source terminal and the gate terminal controls the current flow between source and drain as well as the conductivity of the conduction channel. For a constant source-gate voltage, exposure to gaseous analytes can affect the conductivity of the conduction channel [6,13]. Notably, the main difference between chemiresistors and FET sensors is the ability of the latter to provide not only current variations but also threshold voltage changes upon analyte exposure [142].

Depending on the gaseous analyte-type and the carrier type of the channel material (holes/electrons), the charge carrier concentration of the semiconducting channel material can be changed upon device exposure. In the case of an n-channel FET, oxidizing gas exposure reduces the majority carriers of the channel region thus decreasing the current flow. In the case of exposure to a reducing gas or of a FET with a p-type channel exposed to an oxidizing gas, the current flowing through the channel increases. Measurement of current variations permits the detection as well as the quantification of analytes after appropriate calibration [13]. The applied gate voltage permits control of sensitivity, providing it is set so as to permit the maximum conductance variation [142].

A chemical FET can also possess a gas-selective coating/a series of coatings, between the transistor gate and the analyte. Different chemical modification of the gate allows reaction with different chemical species hence permitting their differentiation [43,143]. For ion sensitive FETs charged species at the sensing interface of the gate, change the polarization of the underlying semiconductor/dielectric interface. Electron conductance through the semiconducting channel is sensitive to gate polarization and the chemical modification of the gate can either attract or repel the semiconductor-charge carriers. Thus, by measuring the source-drain current, the polarization of the sensitive interface is determined [51].

Channel conductivity is also affected by the gas-analyte polarity. Adsorption (molecular gating) of polar molecules on the outer surface of the conducting channel is considered to provoke changes in the electric field. Molecular binding of non-polar molecules can potentially change the density of charged surface states of the functionalized semiconductor surface, due to analyte induced conformational alternations, or can affect the dielectric medium close to the semiconductor surface [6,144].

#### 3.2.1. Selective Applications

Selective FET sensors are commonly developed using nanomaterials for the formation of the conductive channel and used mainly for the detection of simple gases. CNTs (specifically SWCNTs acting as p-semiconductors) have been extensively investigated for CNTFET gas sensors, possessing p-transistor characteristics [7,50]. Pristine CNTs have been used for the detection of the oxidizing gases NO [145] and NO_2_ [145,146,147] and the reducing gas NH_3_ [145,147], as well as for ethanol and benzene detection [147]. Remarkably, the CNTFET sensor developed by Chang et al. exhibited the ability to discriminate the gaseous analytes due to distinguishable temporal response of conductance to gate voltage pulse [147].

2D material-based FETs have been also developed using TMDs or graphene derivatives. WS_2_ n-type semiconducting multilayer nano-flakes have been used for ethanol and NH_3_ detection under illumination [148], while MoS_2_ multilayer-based FETs permit NO [149], NH_3_, and NO_2_ detection [150]. rGO-based FETs have also permitted ethanol [151], NH_3_ and NO_2_ detection achieving low ppb LOD [152], while NH_3_ detection was also reported using NO_2_-dopped graphene [153].

Hybrid materials have also been used in selective FET sensors. 2D graphene/MoS_2_ heterostructures used in flexible and potentially wearable p-type devices, in which graphene replaces the metallic electrodes, have been reported for NO_2_ and NH_3_ detection albeit with lower sensitivity [154]. Combinations of CNTs with other materials for sensing performance improvement are also reported in the literature. Polypyrrole-SWCNTs have been used for NH_3_ detection, achieving increased sensitivity and reduced response/recovery times by controlling film thickness, SWCNTs concentration and annealing temperature [155]. Selective H_2_S detection has been also attempted using Au-decorated SWCNTs, leading to high Au NPs size-dependent sensitivity and lower LOD (≤100 ppb) in comparison to carboxylated SWCNTs (Figure 8) [156]. A lower LOD of 10 ppb and a theoretical LOD of 500 ppt for selective H_2_S detection have been recently achieved using a FET based on Au NPs-functionalized ZnO NWs [157]. Au NPs of 1, 3, 5, 7, and 10 nm were tested, with those of 7 nm leading to the maximum interaction between the analyte and the sensing film.

#### 3.2.2. Cross-Reactive Applications

NWs-FET sensors comprise the most common type used for cross-reactive VOCs detection, including FETs based on SiNWs and MOS NWs (e.g., SnO_2_, ZnO, In_2_O_3_) [63,142]. Using pristine SiNWs, it is possible to discriminate VOCs with high dielectric constants, such as methanol, ethanol, and 2-propanol, with 96% accuracy and acetone, ethanol, and water with 100% accuracy by using pattern recognition methods [158]. As far as MOS NWs are concerned, n-type MOS have been mainly studied. SnO_2_ NWs, e.g., have been used for acetone, ethanol, and methyl ethyl ketone discrimination; the modulation of gate voltage as well as of the operating temperature, permitted the adjustment of sensor response and selectivity [159]. p-Type MOS NWs are also reported to achieve effective VOC detection, such as CuO NWs for NO_2_ and ethanol detection, with ethanol response being reversible by temperature increase due to oxidation towards CO_2_ and water and electron transfer from water to CuO NWs [160]. Complementary-MOS based sensors have also been investigated, along with pattern recognition methods, especially for acetone, acetic acid ethanol, propanol, butanol, and hexanol discrimination [161]. In this case, only one sensor can be used, rather than a sensor array, as selectivity can be achieved by alternating the drain-source and gate potential and without any further modification [161]. Concerning FET arrays based on different nanomaterials, n-type semiconducting In_2_O_3_, SnO_2_, and ZnO NWs combined with a SWCNT-FET have successfully discriminated H_2_, NO_2_, and ethanol [162]. The 4-sensor array was tested at both 200 °C and RT as well as at different analyte concentrations, while SWCNT-FET incorporation improved ethanol and H_2_ overlapping [162].

Remarkably, molecular functionalization has been reported, specifically by Haick et al., for particularly promising FET applications. Molecularly functionalized random CNTs networks have been used for nonpolar and polar VOCs detection. As representative LC biomarkers, decane and 1,2,4-trimethylbenzene have been effectively detected by molecularly modified CNTs [163]. Notably, it has been observed that CNT-functionalization determined the semiconducting character of the material, with tricosane-CNTs leading to a p-type and pentadecane/dioctyl phthalate-CNTs to a n-type behavior thus affecting signal responses [163]. More interestingly, molecularly functionalized SiNWs have been used for cross-reactive FET sensors, targeting exhaled volatile organic biomarkers not only in synthetic but also in real-world breath samples; the discrimination of GCa patients from HC [125] as well as the discrimination of patients with asthma/COPD, LC and GCa (Figure 9) with remarkably increased accuracies [43] (Table 2) are noteworthy examples. As is to be expected, chemical functionalization of different nanomaterials allows for the detection of both polar and non-polar VOCs.

### 3.3. Electrochemical Sensors

Electrochemical sensors are divided into potentiometric (voltage measurement), amperometric (electric current measurement), or conductometric (conductivity or resistivity measurement) [143]. Analyte detection occurs on appropriate electrodes on which a chemical reaction (oxidation or reduction) takes place. Electrochemical sensors typically consist of a sensing (working) electrode and a counter electrode separated by a thin layer of electrolyte [51]. The sensing electrode, on which the reaction occurs, is characterized by high surface-to-volume ratio (for signal enhancement) and is composed of catalytic materials, e.g., platinum, palladium, or carbon-coated metals [143] that are specific for the desired analyte [51]. Analyte-electrode reaction generates a sufficient electrical signal [51] measured with respect to the counter electrode [143]. A schematic representation of electrochemical sensors is presented in Figure 4.

#### 3.3.1. Selective Applications

Electrochemical sensors have mainly been used for the selective detection of gas biomarkers. To this end, conventional, polymeric and hybrid materials have been studied. For example, Prussian Blue electro catalyst-modified carbon electrodes on wearable, paper-based sensor have been developed for H_2_O_2_ detection which serves as a lung-disease biomarker [164]. As an example of polymeric sensing films, cylindrical nano-porous semiconducting polymers have permitted NH_3_ detection in ppb levels via a redox reaction [165]. Hybrid materials, especially the ones containing 2D nanosheets, are also reported in the literature for sensing oxygen-compounds. Nonanal detection for instance has been achieved using SnO_2_ nanosheets decorated with SnO_2_ NPs as well as noble metal catalysts (Pt, Au, and Pd), aiming at LC early-diagnosis [59]. Solid proton-conducting electrolyte based on sulfonic acid co-functionalized cellulose NFs and GO nanosheets, has been developed for ethanol detection via oxidation with a LOD of 25 ppm [166]. More recently, Au NPs-decorated MoS_2_ nano-flakes were used for oxygen-based VOCs detection such as the diabetes biomarker acetone, with sensor responsiveness and selectivity being increased due to electron-donation from Au NPs to MoS_2_ (Figure 10) [69].

#### 3.3.2. Cross-Reactive Applications

Cross-reactive electrochemical sensors have been reported for the in vivo/in vitro diagnosis of diabetes, LC [126], and GCa [137] (Table 2); such devices are based on polymers [126] and nanomaterials (e.g., MNPs, CNTs, SiNWs, graphene derivatives, and their combinations). Remarkably, ultrahigh sensitivity for two GCa biomarkers (Table 2) has been achieved using an Au-Ag NPs-MW CNTs glass carbon electrode, due to the high surface area of both MWCNTs (Au-Ag NPs adsorption enhancement) and Au-Ag NPs (electron-transfer acceleration) [137]. The synergistic catalytic activity of the bimetallic NPs on the other hand enhances the selectivity [137]. Notably, electrochemical sensors are conventionally constrained in detecting electrically inert simple aromatic compounds and hydrocarbons, since the target analyte should be electrochemically active [51]. More recently, the detection of cyclohexane along with formaldehyde, has been reported using a Si NW-rGO sensing film, due to cyclohexane-oxidation catalysis by rGO [106].

### 3.4. Piezoelectric Sensors

In general, piezoelectric materials produce voltage due to the application of mechanical stress and vice versa [51]. Piezoelectric sensors, by definition sensitive to mechanical stress [15], are often used as mass-sensitive sensors [36]. Acoustic wave devices are used in piezoelectric sensors, also called mass, gravimetric, or microbalance sensors. An oscillating circuit is used for the generation of acoustic waves, allowing the piezoelectric crystal to resonate [143]. The most important categories of piezoelectric gas sensors are QCM and SAW [40].

#### 3.4.1. Quartz Crystal Microbalance (QCM) Sensors

QCM sensors possess quartz crystal resonators functionalized with different appropriate sensing elements (e.g., metalloporphyrins [131,132], sensitive polymers, MOS, or nanomaterials) [6,40,167]. The acoustic wave propagates through the bulk of the crystal perpendicularly or parallel to the surface [51]. When a gas is absorbed on the sensitive surface of a crystal the mass changes thus alternating the resonance frequency [167]. Typically, mass increase results in a decrease in the oscillation frequency of the resonator [143] which comprises the measured physical quantity [167]. The sensing mechanism of QCM sensors is presented in Figure 11a.

##### Selective Applications

QCM sensors are used extensively for selective gas detection and are potentially applicable in breath analysis. As an example of MOS based sensors, ZnO has been used in the form of NWs [168] and vertically-aligned NRs [169] for NH_3_ detection, permitting the development of reproducible and stable systems in both cases as well as increased selectivity against liquefied petroleum gas, N_2_O, CO, NO_2_, and CO_2_ in the case of NRs. Notably, VOC detection with MOS-QCM sensors is feasible at RT (despite the use of MOS) since it is directly connected to mass-changes [169]. Concerning polymeric materials, for the selective detection of NH_3_ [170] and formaldehyde [171], the use of polymeric NFs based on poly(acrylic acid)/poly(vinyl alcohol) (PAA/PVA) [170] and polyethylenimine (PEI)/PVA [171], respectively, has been demonstrated. More recently, a Si NPs-containing methacrylic acid-based MIP-composite was reported to selectively detect hexanal as LC biomarker in the presence of trimethyl amine, NH_3_, ethanol, acetone, acetic acid, and diethyl ether, due to analyte binding through hydrogen bonding formation [74].

##### Cross-Reactive Applications

In the case of cross-reactive applications, hybrid and especially polymeric/macrocyclic sensing materials are investigated. TiO_2_NPs-decorated MWCNTs and cobalt (II) phthalocyanine-silica, for instance, have been used as sensing elements in a 3 sensor array, along with a bare QCM sensor; this was used for simultaneous acetone and NO detection, which serve as diabetes and asthma biomarkers, respectively [139]. No response was observed by the bare sensor, while analyte adsorption to the modified sensors is attributed to analyte-sensing film coordination interactions [139]. Concerning polymeric materials, eight different polythiophene derivatives have achieved acetic acid, toluene, acetone, *p*-xylene, ethanol, 1-octanol, acetonitrile, and water discrimination based on VOC-polarity, with a LOD at ppm levels; prediction of toluene concentration in mixtures with ethanol was also permitted [173]. Macrocyclic calixarene derivatives, on the other hand, have achieved the detection of 16 VOCs belonging to ketones, alcohols, aromatic, and chlorinated compounds, at ppm levels, due to “host–guest” interactions. Sensor sensitivity and selectivity were dependent on the number and functional groups of calixarene derivatives [174]. In a more recent application, 13 different MIPs of different compositions and cavity structures were studied for the detection of hexanal, nonanal, and benzaldehyde, which are common cancer biomarkers. A combination of five sensors led to the most effective VOC discrimination while sensor-array performance was found to be dependent on both molecular imprinting and matrix effect [73].

Metalloporphyrin-based cross-reactive QCM sensors seem as the most promising QCM sensors (Table 2) while they have been also applied in real breath samples [130,131,132,133,134]. Such e-Noses have been examined for asthma [131], COPD [130], halitosis [135], LC [132,133,175], and tuberculosis [134] diagnosis with particularly promising diagnostic performance. Similarly, an anthocyanin-based QCM sensor called BIONOTE e-Nose has been developed by Santonico et al. [176] and was recently studied for use in asthma diagnosis of children, with the aim to assess the within and between-day repeatability of obtained measurements; values similar to those of conventional methods were achieved (Table 2) [136].

#### 3.4.2. Surface Acoustic Wave (SAW) Sensors

In *SAW* gas sensors, the acoustic wave propagates only parallel to the surface of the piezoelectric crystal, penetrating about one acoustic wavelength in depth into the crystal. Motion at the surface is both parallel and perpendicular to it [51]. Crystal surface is modified with a chemically selective layer [40,143]. Exposure to the analyte affects the propagation waves [143], as the mass (acoustic field of the SAW) and/or the electrical conductivity (electric field of the SAW, associated with the acoustic field) of the chemical interface change [15]. As a result, a change is induced in the propagation frequency of the SAW, which can then be measured [6,40]. The sensing mechanism is presented in Figure 11b.

##### Selective Applications

Similar to QCM sensors, MOS including ZnO, SiO_2_, TiO_2_, Co_3_O_4_, WO_3_, and other combinations [7,172], have been used extensively for SAW devices and most commonly for selective VOCs detection. For example, ethanol detection has been achieved at 300 °C using a YX LiTaO_3_-based SAW sensor modified with a ZnO intermediate layer of 1.2 μm and a WO_3_ sensing layer of 150 nm [177]; porous ZnO-SiO_2_ bilayer nano-films have been used for NH_3_ detection, with SAW sensor sensitivity being better than the one for single layers and was found to be dependent on bi-film conductivity [178]. In the latter system, it was observed that increased ZnO film thickness led to larger absolute sensor response but also to larger response and recovery times; the study concluded that 60 nm of ZnO led to the best sensing performance for 30 ppm NH_3_ [178]. The detection of NH_3_ concentration levels that were lower than 1 ppm has been also attempted by the same group in the presence of H_2_, CO, CH_4_, H_2_S, and ethanol using SiO_2_-TiO_2_ films of 200 nm that exhibited remarkable system selectivity, stability, and reproducibility [179].

##### Cross-Reactive Applications

Non-functionalized [180,181] or polymer-functionalized [129] SAW sensors have been combined with GC columns as detectors for the development of point-of-care diagnostic systems. BC [180], tuberculosis [181], and LC [129] patients for instance, have been effectively diagnosed with an accuracy of 79%, 84%, and 80%, respectively. SAW sensors have been also used for the detection of both polar and non-polar VOCs, after their modification with appropriate sensing films [6].

Cross-reactive SAW sensor arrays with polymeric coatings of different composition [182] or thickness [183] have been reported; SAW sensors coated with eight different polymeric films have achieved effective discrimination of chloroform, octane and xylene vapors as shown by radar plots, maintaining their performance for a period of 3 years [182]. Interestingly, 3 polyisobutylene films of varying thickness, namely 10, 50, and 100 nm, allowed for chloroform, chlorobenzene, o-dichlorobenzene, heptane, toluene, hexane, and octane discrimination by analyzing the transient responses characterized by analyte-specific kinetics variability (different stages of equilibrium attainment, for different coating thickness) [183]. Such polymer-based SAW sensors hold a great promise for potential breath analysis applications. CNTs, SWCNTs, and MWCNTs dispersed in ethanol or toluene for example, have been separately tested for ethanol, toluene, and ethyl acetate sensing at RT with a LOD of 1 ppm [184]. The main advantage of CNTs-modified sensors is the enhanced SAW sensor-sensitivity, due to the ability to sense variations not only in mass but also in conductivity [172].

Among hybrid materials, MWCNTs combined with other materials have achieved selectivity enhancement. Polyepichlorohydrin and polyurethane were combined with different MWCNTs percentages so as to develop a 4 sensor-array, for toluene and octane detection and differentiation using polar plots [185]. Notably, no response was observed for gases such as H_2_, NH_3_, NO_2_, and CO, while toluene adsorption was far more enhanced than that of octane [185]. In a different publication, MWCNTS combined with nano-sized CeO_2_ (100:1) have formed a semiconducting composite that achieved acetone and ethanol detection, with higher sensitivity for the former [186]. As far as MNPs-based hybrid materials are concerned, thiol containing calix4arene-modified AuNRs and Ag nano-cubes (NCs) have been recently used in a 5-sensors array, for the detection of polar and non-polar VOCs in low ppm levels including the LC biomarker toluene. Chloroform and toluene were detected with greater sensitivity by the modified AuNRs and AgNCs respectively when compared to unmodified NPs (Table 2) [49]. Calix4arene modification enhanced sensitivity largely due to the increase of the surface interaction area as well as the arrangement of the macrocyclic ligands (Figure 12) [49]. The use of pristine or Au NPs-functionalized zeolitic-imidazole-framework nanocrystals (ZIF-8 and ZIF-67) in a 4-sensor array for the discrimination of diabetes biomarkers (acetone, ethanol, and NH_3_) via PCA is another example of promising hybrid materials [138] (Table 2). Molecularly functionalized MOS-NPs are also reported in the literature. Using a layer of amino-terminated iron oxide NPs for instance, the VOCs butanol, isopropanol, toluene, and xylene were detected with a low LOD of 1, 12, 3, and 0.5 ppm, respectively, thus developing an effective sensing device of lower cost in comparison to MNPs-based sensing systems [187]. Iron oxide NPs have been also used in combination with polymers. Fe_3_O_4_ NPs of varying diameter (7, 13, or 50 nm for 0.4 mg NPs/mL polymer solution) and concentration (0.2, 0.4, or 0.8 mg NPs/mL polymer solution for NPs 50 nm), have been embedded in PEI [52]; by employing a PEI functionalized sensor, a 6-sensor array was developed for the effective detection of methanol, ethanol, and toluene while sensitivity increased for smaller NP-diameters and greater concentrations. Remarkably, NPs of 7 nm permitted ethanol detection with a LOD five times better (65 ppm) than solely PEI-based sensors [52].

In comparison to QCM sensors, SAW based sensors are generally characterized by higher sensitivity, while the potential for surface modification is expanded. On the other hand, it is worth noting that in both cases apart from the sensing film composition that determines sensor selectivity, the high surface area of the nanostructures comprises the fundamental factor that enhances sensitivity due to the creation of more adsorption sites (defects) [52,169,171,187]. In general, piezoelectric gas sensors, investigated primarily in synthetic samples, are characterized by increased sensitivity, small response time and low-powered operation. However, the low signal-to-noise ratio and the requirement for complex electronic circuits, may render this sensor type less enticing for the development of efficient e-Noses [40].

### 3.5. Optical Sensors

Optical gas sensors detect analytes by measurable changes of absorption, luminescence, scattering, reflectivity, refractive index or optical path length [40], due to the interaction of the radiation with the desired gas or a selective layer [143]. In the first two cases, light intensity or wavelength are measured [40].

#### 3.5.1. Optical Fiber-Based Sensors

Optical fibers possessing a chemical reagent (e.g., chemical dye) or a sorbent phase (e.g., polymeric film) as a reactive layer, are commonly used as transduction elements in optical gas sensors [6,143] specifically for VOC detection [6]. Upon vapor exposure optical or structural changes of the reactive layer alter the effective index and hence the light transmission properties of the fiber [6].

##### Selective Applications

Nanomaterial-based hybrid materials have been used for optical fiber modification as sensing films for selective VOC detection. The use of graphene, commonly combined with MNPs [188,189], is a representative example. Pt NPs-GO [188] and Ag NPs-GO [189] functionalized optical fibers have been developed for efficient and selective NH_3_ detection (NH_3_ is a common biomarker for renal and liver diseases as well as *Helicobacter pylori* infection) [7]. Remarkably, Pt NP-functionalization increased the sensitivity in comparison to pristine GO [188], while the concentration of Ag NPs was inversely correlated with sensitivity [189]; this reveals the benefits as well as the vulnerability of the synergistic effects of hybrid materials. An additional hybrid material, namely a thin film of poly (allylamine hydrochloride) and Si NPs infused with tetrakis (4-sulfophenyl) porphine, has been used as an optical-fiber coating for the selective detection of methanol in the presence of water and other alcoholic vapors [190].

##### Cross-Reactive Applications

Standalone polymers or polymers combined with nanomaterials have also been used for cross-reactive applications. PMMA-based fibers functionalized with nano-crystalline bismuth oxide-clad have effectively detected NH_3_, ethanol, methanol, and acetone, exhibiting increased selectivity towards methanol [191]. Detection of hydrocarbons and aromatic compounds in real breath samples of HC, as potential biomarkers of various diseases, has been also achieved using a polymethyl (3,3,3-trifluoropropyl) siloxane-coated optical fiber; the device features a low LOD posing as an attractive alternative for disease diagnosis (Table 2) [128]. Despite their extensive investigation in the field of breath analysis for VOCs detection, such devices are scarcely studied for real sample experiments [6].

#### 3.5.2. Colorimetric

Colorimetric sensors, usually classified as a sub-group of optical sensors [51], are based on environmentally dependent color changes [15]. Chemo-responsive indicators are able to chemically react and change color in a distinctive way upon exposure to different gas analytes [6], thus permitting analyte identification. In this case the response upon analyte exposure is based not on the physical properties but the chemical reactivity of the indicators [15].

##### Selective and Cross-Reactive Applications

Selective applications of colorimetric sensors have scarcely been studied. Lead acetate (PbAc_2_) NPs anchored to polyacrylonitrile nanofibers (NFs) comprise an example of a selective sensing element that was investigated for H_2_S colorimetric detection (a halitosis biomarker); a LOD of 400 ppb which is far lower than the 5 ppm of PbAc_2_ paper tests, was achieved [192]. Concerning cross-reactive sensing systems, there are five possible categories to which an indicator may belong to: pH responsive, (Brønsted acidity/basicity); metal-ion-containing dyes (Lewis basicity—electron pair donation); redox-responsive metal salts; nucleophilic indicators (responsive to electrophilic analytes) and dyes with large permanent dipoles (e.g., solvatochromic dyes) [6]. Such optical sensors have been extensively investigated in the field of breath analysis, mainly for LC diagnosis, using both synthetic [140,193] and real-world breath samples [127,194]. Hou et al. have achieved to discriminate at first 4 (Table 2) [140] and then 20 [193] LC-related VOCs in two separate studies, with accuracies of 100% and 90%, respectively. Mazzone et al. achieved to differentiate LC patients and HC, with moderate accuracies [127,194], as well as LC patients of different histologies with a higher accuracy percentage (Table 2) [127]. The main challenges for such sensors include low sensitivity, high LOD and low response times. Furthermore, their irreversible operation renders them disposable (single-use tests), which is something that should also be considered [7,167]. Nanomaterials may be also used for enhanced sensing properties. For the detection of VOCs that have been identified as LC biomarkers the use of AuNRs-modified metalloporphyrins has proven to protect the device from photo-degradation, to provide good repeatability, increased long-term sensor stability, and increased shelf-life (Figure 13) [141].

#### 3.5.3. Localized Surface Plasmon Resonance (LSPR)-Based Sensors

Non-conventional chromophores, such as MNPs, have also been investigated in optical sensors, due to their interesting optical properties. LSPR-based gas sensors [36,40] are based on the dependence of the LSPR properties of the MNPs from the refractive index of the dielectric environment surrounding the NP [36] (i.e., coating, surrounding medium, supportive substrate [51]). Refractive index alternation of the medium changes the wavelength of the incident light [36]. The main advantage of MNPs is the fact that the extinction coefficients are several orders of magnitude higher than those of conventional dyes, in the visible spectrum, allowing higher sensitivity and lower LOD for the desired analyte [51].

Optical detection of VOCs using various compositions and shapes of MNPs has also been reported. Au NPs, Ag NPs, and Au NSs have been used for chlorobenzene, *m*-xylene, pentanol, toluene, and octane cross-reactive detection [195]. Hybrid materials containing MNPs have been used as well. Selective polymer-coated Au nano-islands, for instance, have effectively detected a-pinene [196].

#### 3.5.4. Surface Enhanced Raman Spectroscopy (SERS)-Based Sensors

Another, more widespread optical-sensor application of the LSPR phenomenon in MNPs, is SERS [36], i.e., the enhancement of Raman signals due to the LSPR phenomenon. SERS is a vibrational spectroscopic technique [197] that permits single-molecule detection, and has been investigated among others for VOCs detection [198].

Functionalization of different MNPs with appropriate molecules has been reported for selective VOC-detection. Aldehydes detection as LC biomarkers by SERS-based sensors seems promising [198,199]. Dendritic Ag nanocrystals coated with the aldehyde-selective probe molecule *p*-aminothiophenol have achieved low LOD (ppb range) in the presence of confounding LC biomarkers (hydrocarbons, alcohols, ketones, esters, nitrogen, and aromatic compounds) [199]. NO, on the other hand, serves as a biomarker for asthma [36], hypertension, arteriosclerosis, diabetes, and rheumatoid arthritis [197]. NO detection has been achieved by *o*-Phenylenediamine-modified Au NPs which have permitted selective chemical reaction between functionality moieties and NO, leading to nano-probe SERS variations; such sensors have achieved a LOD of 1.7 × 10^−7^ M, in the presence of H_2_S and CO [197].

As it can be observed, SERS-based sensors have been developed for the selective detection of a specific VOC or classes of VOCs, rather than cross-reactive detection. Notably, optical sensors are not preferred for e-Nose development, mainly due to their size and complex signal conditioning [40].

On Table 2 representative examples for different types of sensor-arrays, using conventional materials or nanomaterials, are presented. Sensitivity, selectivity, and discriminant accuracy of the sensor arrays highlight their promising application as diagnostic tools. It can be observed that the incorporation of nanomaterials in the sensing element ameliorates the sensing performance (i.e., sensitivity, LOD), while appropriate modification permits the desired cross-selectivity. Sufficient LOD, similar to the usual concentration of VOCs in exhaled breath, is achieved by all sensor types. As far as MOS-based sensors are concerned, it is apparent that their main drawback towards all the other categories is their increased operating temperature. Chemiresistors are probably the most investigated gas sensor type for cross-reactive systems, incorporating different (nano)materials, being particularly attractive for the diagnosis of a wide range of diseases. Remarkably, most of the branded e-Noses that have been developed and tested for breath analysis applications, presented on Table 3, appertain to chemiresistors (AeoNose, Cyranose 320, SpiroNose, DiagNose, and Common Invent e-Nose). Another noteworthy information presented on Table 2 is the variety of data analysis and machine learning techniques used among the different studies, in the absence of which the discrimination of subjects would be infeasible.

## 4. Differential Diagnosis and Disease Phenotyping and Staging in Breath Analysis

As can be observed on Table 2, exhaled VOCs have been used not only for disease diagnosis in comparison to healthy subjects, but also for the discrimination between different diseases (e.g., GCa, LC, and asthma/COPD [43]) or even for the discrimination between the different stages of a particular disease (e.g., CKD stages [113]). One of the main prerequisites for the development of clinically applicable diagnostic tests is the effective discrimination between different diseases with similar symptoms and biochemical pathways [200]. The uncertainty in the differentiation of patients with distinct diseases comprises one of the main drawbacks for studies distinguishing a specific disease from HC [201]. Thus, the use of breath analysis for differential diagnosis as well as disease staging or phenotyping, using either analytical techniques or sensing devices (Table 4), attracted significant research interest over the last years. As expected, the contribution of nanomaterials is of great importance with many recent publications of nanomaterial-based sensors focusing on disease differentiation, staging or phenotyping, rather than the simple discrimination between patients and HC (Table 4).

### 4.1. The Case for Lung Diseases

Chronic and acute lung diseases such as asthma, COPD, idiopathic pulmonary fibrosis (IPF), LC, mesothelioma, and sarcoidosis have been connected with similar metabolic alternations [202]. Especially asthma and COPD are also characterized by similar symptoms [37] with COPD being commonly underdiagnosed or diagnosed at late stages [203]. Concerning LC, no symptoms are expressed in early stages [204] while disease manifestation is limited to non-specific symptoms [204] including cough, short breath, chest pain, and weight loss [37]. Disease phenotyping, on the other hand, is mandatory in some cases. Asthma subtypes such as eosinophilic, neutrophilic, mixed granulocytic, and paucigranulocytic asthma [205] are characterized by similar symptoms while different treatment is required [206]. Similarly, immunosuppressive, antifibrotic, or a combination of medications may be needed for fibrotic interstitial lung diseases (ILDs), depending on the respective phenotype (inflammatory, more fibrotic, or combination) [207]. Thus, reliable phenotyping is needed for appropriate medication to be administered [33,206,207]. LC is subdivided into different categories with different clinical characteristics as well. Small cell LC (SCLC), with 20–25% percentage of occurrence [208], is characterized by increased metabolic and proliferation rates compared to other cancer cells [209], while non-small cell LC (NSCLC) accounts for 70–75% of LC cases and is subdivided into the smoking-related [37,210] squamous cell carcinoma (SCC) [208] and the non-squamous cell carcinomas [37] including adenocarcinomas (ADC) (minor smoking correlation) and large-cell carcinoma (LCC) [208]. Consequently, the accurate discrimination of different lung diseases and subtypes of a lung disease, especially using breath analysis of exhaled VOCs, is of particular importance.

The use of GC-MS has rendered lung disease differentiation, phenotyping and staging feasible in many cases. To be more specific, LC discrimination from patients with other lung diseases has been investigated in several studies. In an attempt to discriminate between NSCLC, COPD, and HC patients, by taking smoking habits into account, 4 VOCs were identified (in varying concentrations) for NSCLC and COPD [211]. In another study, Wang et al. attempted to discriminate LC from COPD, asthma, pneumonia, pulmonary embolism and benign lung tumor patients; however, the selected 10 VOCs could not discriminate accurately between the two groups, implying their potential confounding role during LC-biomarkers determination [18]. Koureas et al. have also attempted to discriminate LC from other respiratory diseases, using 19 distinctive VOCs, based on the underlying disease mechanisms (targeted method); only the discrimination of LC patients from HC, using ethylbenzene, toluene, styrene, 2- and 1-propanol was achieved [200]. However, in a more recent study of the same group, the discrimination of LC patients from patients suffering from sarcoidosis, hypersensitivity pneumonitis, interstitial lung diseases or pulmonary infections was achieved with an increased accuracy of 75.3%; the 29 VOCs were selected following a hypothesis-generating non-targeted strategy [212]. In a different study, LC was accurately distinguished from pulmonary non-malignant diseases (PNMD; COPD, pulmonary tuberculosis, asthma) using 10 VOCs while 5 were selected as characteristic of LC in contrast to both PNMD and HC [213].

The differentiation of LC patients from patients with benign pulmonary nodules (BPN) has also been extensively reported. Apart from a study by Wang et al. [18] which included patients with benign lung tumors, Fu et al. have investigated the respective discriminant ability of carbonyl VOCs [214,215,216]. Four carbonyl VOCs, captured by a silicon micro-reactor, were found to present increased concentration in LC patients when compared to BPN patients and HC [214]. In a subsequent study, the same group achieved the differentiation of both early and III, IV stage LC patients from BPN patients, with high sensitivity (83%) and particularly increased specificity (74%) in comparison to positron emission tomography (90% and 39%, respectively) [215], while 6 carbonyl VOCs have permitted a classification accuracy of 89% of LC vs. BPN patients [216]. More recently, Chen et al. identified 19 VOCs able to distinguish not only LC and BPN patients (this with an accuracy of 80.9%) but also early-stage LC patients from BPN (with an accuracy of 75.6%), being remarkably promising for early LC diagnosis [204].

LC histology and staging characterization using analytical methods is another important target of this research field. It was reported that 1-butanol, 3-hydroxy-2-butanone [9], as well as 4-hydroxyhexenal [214], can differentiate SCC from ADC patients with the former being decreased for SCC in contrast to the other 2 VOCs. Similarly, SCLC and NSCLC can be potentially distinguished from 4-hydroxynonenal and C_5_H_10_O [214]. Hexanal has also been found in higher concentrations for SCLC patients compared to NSCLC, potentially due to increased metabolic rates [209]; Chen et al. have achieved NSCLC and SCLC differentiation, with an accuracy of 93.9%, using a pattern of 20 VOCs [204]. Concerning LC staging (I, II, III, or IV), a pattern of 19 VOCs was used to distinguish between early (I, II) and advanced LC stages (III, IV) with 82.7% accuracy [204] while Fu et al. demonstrated that exhaled 2-butanone concentration is significantly different between stages I and II–IV [214].

Apart from LC, other lung diseases are also studied for accurate diagnosis. A series of studies have focused on asthma phenotyping using GC-MS. Brinkman et al. identified 3 VOCs significantly correlated with sputum eosinophils [90], while Ibrahim et al. identified VOC-patterns differentiating eosinophilic from non-eosinophilic (6 VOCs) and neutrophilic from non-neutrophilic (7 VOCs) [217]. Recently, Schleich et al. identified 4 VOCs discriminating eosinophilic from neutrophilic, eosinophilic from paucigranulocytic and neutrophilic from paucigranulocytic asthma, with accuracy similar to blood eosinophils and FeNO tests [205]. In a more recent study, the same group used two-dimensional GC-high resolution-time-of-flight-MS, selected ion flow tube mass spectrometry (SIFT-MS), 10 VOCs, and 9 ion channels so as to achieve asthma phenotyping with an accuracy of 75% [206]. COPD phenotyping and staging has also been attempted using analytical techniques. Fens et al. identified 8 eosinophils- and 17 neutrophils-related VOCs, with only one VOC overlapping between the two subgroups. More VOCs were related with cell counts for Global Initiative for Obstructive Lung Disease (GOLD) stage II, in comparison to GOLD stage I [120]. In another study, 11 COPD patients with >1% and 6 with >2% eosinophil count were discriminated from non-eosinophilics (<1% and <2% eosinophil count, respectively) with accuracies of 79% and 92% [218]. Exacerbation prediction of both asthmatic children [219,220,221] and adults [90,217] as well as COPD patients [222] also comprises a subject of study.

Following the promising applications of analytical methods that highlight the potential capabilities of breath analysis in phenotyping, staging and differential diagnosis of lung diseases, sensing devices have also been used in respective applications with remarkable results. Research interest has focused on LC discrimination from other lung diseases such as COPD and asthma. Cyranose 320 has achieved separation of NSCLC from COPD (GOLD stage I–III) with an accuracy of 85%, in an article by Dragonieri et al. [223]. Tirzīte et al. have used this e-Nose to effectively discriminate, not only LC patients from COPD, asthma, pneumonia, pulmonary embolism, benign lung tumor patients, and HC, with 87.3% accuracy, but also between LC patients, COPD patients, LC patients suffering also from COPD and HC with 77.4% accuracy, and totally correct classification of the 79 LC/COPD patients [224]. In a more recent study, the same group discriminated LC from patients with non-malignant lung diseases as well as bronchiectasis, tuberculosis, and HC by taking into account smoking habits. An overall sensitivity and specificity of 95.8% and 92.3% for smokers and 96.2% and 90.6% for non-smokers, respectively, was observed using Cyranose 320 [225]. More recently, the same e-Nose was used by Rodriguez et al. for the discrimination of COPD from LC and BC, achieving an overall correct classification of 91.35% while LC correct classification in relation to COPD was equal to 96.47% [201]. Interestingly, the contribution of the 32 sensors in the discrimination was also assessed [201]. Tor Vergata e-Nose has been used effectively for discriminating LC patients from COPD, Interstitial lung disease, Pleurisy and Bronchitis patients, with a sensitivity of 89.3% for LC patients [175]. In a particularly promising study, SpiroNose discriminated LC, asthma, COPD, and HC, with the respective accuracy values presented in Figure 14a (68–88%) [91]. The applicability of breath sampling and analysis was tested as the collection of asthma breath samples at two different sites led to similar results (Figure 14b) [91]. In the field of non-commercial and self-developed sensors, Tan et al. have attempted to develop a cross-reactive alkane-based chemiresistor combining carbon powder and tetracosane, achieving not only high affinity for alkanes and low sensitivity for polar VOCs (water, ethanol, ethanal) but also effective differentiation of 12 LC patient from 13 HC and 12 COPD patients [108]. In a more recent study, researchers attempted to differentiate LC, COPD, and asthma patients from HC, using an array of 8 sensors of 4 different types (MOS, electrochemical, hot wire, and catalytic type). The array achieved accuracy between 76.9–84.75%, using different machine learning methods [226]. Accuracy values were greater for LC and COPD prediction; however, the maximum accuracy value of 84.75% was attained using kernel principal component analysis—extreme gradient boosting (KPCA-XGBoost), which indicates excellent discriminatory capability for LC and COPD patients [226]. Similarly, using an array of 11 sensors of 4 different types (namely MOS, electrochemical, hot wire, and catalytic type), Liu et al. differentiated non-smoking LC and COPD patients, with the best discriminatory accuracy (96%) being achieved using the same machine learning technique [227]. The discrimination of LC from asthma and COPD patients was also achieved by Haick’s group with particularly high classification accuracies [43]. Early-stage LC discrimination from BPN has been reported by Haick et al. using an array of 40 chemiresistors based on MCNPs (Au NPs) and molecularly-coated SWCNTs, achieving an accuracy of 87%. Considering that the required treatment may change in the occurrence of genetic alternations, the differentiation of patients with and without epidermal growth factor receptor (*EGFR*) mutation was also attempted with an accuracy of 83% [228].

**Table 4 sensors-22-01238-t004:** Sensing devices used for differential diagnosis, staging, and phenotyping of different categories of diseases.

Sensor	Diseases/Phenotypes/Stages	Subjects	Classifier	Results	Ref.
**Differential diagnosis**					
**Cyranose 320**	NSCLC vs. COPD (GOLD stage I-III)	10 NSCLC, 10 COPD	PCA, CDA	85% acc.	[223]
	LC vs. non-cancer (COPD, asthma, pneumonia, pulmonary embolism, BPN)	165 LC, 91 non-cancer	SVM	87.3% acc.	[224]
	LC vs. COPD vs. LC/COPD vs. HC	63 LC, 15 COPD, 79 both, 78 HC		77.4% acc., 100% accurate LC/COPD classification	
	LC vs. non-cancer (COPD, asthma, pneumonia, pulmonary embolism, bronchiectasis, BPN, TB)	252 LC, 223 non-cancer	LRA	Sens.: 95.8% (S), 96.2% (NS) Spec.: 92.3% (S) 90.6% (NS)	[225]
	Asthma vs. COPD	20 asthma, 30 COPD	PCA, CDA	96% acc. Within/between day repeatability, reproducibility of e-Noses	[229]
	Fixed and classic asthma vs. COPD (GOLD stages II-III)	21 fixed asthma, 40 COPD	PCA, CDA	88% acc., 85% sens., 90% spec.	[230]
		39 classic asthma, 40 COPD		83% acc., 91% sens., 90% spec.	
	IPF vs. COPD	32 IPF, 33 COPD	PCA, CDA	80% cross-validated acc., Wider patient cohorts and inclusion of more comorbidities needed	[231]
	COPD vs. LC vs. BC	50 COPD, 30 LC, 50 BC	PCA, CDA, CAP	Correct classification values: LC vs. COPD 96.47%, LC vs. BC 93.05%, BC vs. COPD 100%, COPD vs. LC vs. BC 91.35%	[201]
	Bronchial vs. Laryngeal SCC (advanced)	10 bronchial, 10 laryngeal	JMP Pro	10% misclassification, 100% sens., 80% spec.	[232]
	AD vs. PD vs. HC	18 AD, 16 PD, 19 HC	LDA	76.2% sens., 45.8% spec., *p* < 0.0001	[233]
**AeoNose**	ILDs (COP, CTD) vs. COPD, ILDs subgroups (COP, HP, IPF, sarcoidosis, uILD, asbestosis, NSIP, RB-ILD, DIP)	28 COP, 23 COPD	Athena program, *t*-test	AUC 0.77, 75% sens., 71% spec.	[234]
		25 CTD-ILD, 23 COPD		AUC 0.85, 88% sens., 71% spec.	
		174 ILDs, 23 COPD		Less accurate discrimination of ILDs subgroups (e.g., AUC IPF vs. CTD-ILD 0.86, COP vs. CTD-ILD 0.82)	
	Asthma vs. CF	20 asthma (moderate-severe), 13 CF	ANN	AUC 0.90, 89% sens., 77% spec.	[235]
	HNSCC vs. LC	52 HNSCC, 32 LC	ANN	Acc., sens., spec.: 93%, 96%, 88% (best fit), 85%, 85%, 84% (cross-validation)	[236]
	Cancer types	100 HNSCC, 40 bladder, 28 colon cancer	ANN	Acc., sens., spec.: 81%, 79%, 81% HNSCC vs. colon cancer, 84%, 80%, 86% HNSCC vs. bladder cancer, 84%, 88%, 79% Colon vs. bladder cancer	[237]
**SpiroNose**	LC vs. COPD vs. asthma vs. HC	31 LC, 31 COPD, 37 asthma, 45 HC	PCA	Cross-validation values 78–88%, repeatability ↑	[91]
	ILD subgroups:	141 sarcoidosis, 85 IPF, 33 CTD-ILD, 25 HP, 11 IPAF, 10 NSIP	PLS-DA	Acc., sens., spec.: 77%, 75%, 84% IPF vs. HP, 94%, 98%, 85% IPF vs. CTD-ILD, 92%, 92%, 90% IPF vs. NSIP, 89%, 87%, 100% IPF vs. IPAF, 75%, 100%, 67% CTD-ILD vs. IPAF, 98%, 90%, 100% CTC-ILD vs. NSIP, 90%, 94%, 72% HP vs. sarcoidosis, 91%, 92%, 88% (training), 91%, 95%, 79% (validation) IPF vs. non-IPF	[207]
**Chemiresistor-based** **alkane sensor**	LC vs. HC LC vs. COPD	12 LC, 12 COPD, 13 HC	MANOVA	LC: 83.3% sens., 88% spec. Sensor acc no smoke-dependence	[108]
**MOS, electrochemical, hot wire, and catalytic**	LC vs. COPD	48 LC, 52 COPD	8 different	76.9–84.75% acc., 75–81.36% sens., 78.79–88.14 spec. Highest acc. With KPCA-XGBoost	[226]
	LC vs. COPD	33 LC, 28 COPD	PCA-SVM, KPCA-SVM, PCA-XGBoost, KPCA-XGBoost	82.52–96% acc., 78.33–95% sens., 85–96.67% spec. Highest acc. With KPCA-XGBoost	[227]
**Organically-coated AuNPs and SWCNTs based chemiresistor**	LC (I/II) vs. BPN	16 LC, 30 BPN	DFA	87% acc., 75% sens., 93% spec. Low LC sample → careful interpretation	[228]
	BC vs. benign	30 HC, 15 BBT, 13 DCIS, 96 BC	DFA	Acc., sens., spec.: 88.3%, 90.6%, 83.3% BC vs. BBT/HC, 71.2–82%, 62.6–80%, 75.7–82.3% BC vs. BBT, 81.4–84.4%, 83–83.3%, 81–92% BC vs. DCIS	[238]
	Gca vs. OLGIM groups (0-IV)	99 Gca, 155 OLGIM 0, 136 OLGIM I-II, 34 OLGIM III-IV, 53 PUD	DFA	Acc., sens., spec.: 92%, 73%, 98% Gca vs. 0–IV, 84%, 90%, 80% Gca vs. 0, 87%, 97%, 84% Gca vs. 0–II, 90%, 93%, 80% Gca vs. III-IV, 85%, 93%, 80% Gca vs. I–IV, 87%, 87%, 87% Gca vs. PUD	[239]
	Gca vs. benign gastric conditions	37 Gca, 32 ulcers, 61 less severe conditions	DFA	89% sens., 90% spec. 84% sens., 87% spec.	[240]
	ulcer vs. less severe				
	AD vs. PD AD vs. PD vs. HC	15 AD, 30 PD, 12 HC	DFA	AD vs. PD: 84% acc., 80% sens., 87% spec. Feasible overall discrimination, with large PD/HC overlap	[114]
**NA-NOSE**	BC, benign breast conditions, normal mammographs	11 BC, 14 benign, 7 normal mammographs	PCA/ANOVA/Student’s *t*-test, SVM	94% sens., 80% spec. for benign vs. BC and negative mammography, Similar results with both methods	[116]
**MCNPs-based chemiresistor—6 sensors-array**	IBD vs. IBS	71 IBD (35 UC, 36 CD), 26 IBS	ANN	81/88% acc., 92/73% sens., 53/100% spec. (real/artificial)	[111]
	CD vs. UC			75/96% acc., 75/100% sens., 47/93% spec. (real/artificial)	
**Molecularly modified SiNW FET**	Gca vs. LC	40 Gca, 149 LC, 56 asthma/COPD	DFA, ANN	92% acc., 93% LC, and 85% Gca correct classification	[43]
	LC vs. asthma and COPD			89% acc., 92% sens., 80% spec.	
**MOS gas sensor array**	Gca vs. gastric ulcer patients	49 Gca, 30 gastric ulcer	Back-propagation Neural network	93% acc., 94.38% sens., 89.93% spec. Classification acc. Of malignant, benign, normal subjects: 92.54%, 93.17%, 92.49%.	[241]
	OC vs. benign and HC	86 OC, 51 benign, 114 HC	PCA, k-NN	Acc., sens., spec.: 85%, 6%, 84% (cross-validation/strict prediction), 87%, 89%, 86% (prediction/strict prediction), 86%, 84%, 85% (cross-validation/most probable pred.), 100%, 100%, 100% (prediction/most probable pred.)	[242]
	AD vs. PD vs. HC	20 AD, 20 PD, 20 HC	PCA	Effective discrimination of AD vs. PD and HC	[243]
**BIONOTE**	CLD vs. NC-CLD	65 CLD, 39 NC-CLD	PLS-DA, radar plot	Successful discrimination, 16 cirrhotic patients misclassified	[244]
**Commercial (MQ) gas sensors**	CKD vs. diabetes vs. HC high creatinine, HC low creatinine	84 CKD, 24 diabetes, 54 HC high creatinine, 54 HC low creatinine	Radar plots, PCA, SVM, PLS-regression, HCA	PCA: 96.64% of total variance expressed in PC1–3 SVM: 100% correct classification of samples	[245]
**Disease histology/phenotyping**					
**Tor Vergata e-Nose**	SCC vs. ADC	10 SCC, 10 ADC	PLS-DA	75% correct classification	[132]
**24 colorants**	SCC vs. ADC	22 SCC, 50 ADC	LPM	86.4% acc.	[127]
	SCLC vs. NSCLC	9 SCLC, 83 NSCLC		78.1% acc. (moderate)	
**UV-irradiated pristine, Au, Pt, Au/Pt, Ni, Fe-doped WO3NWs**	LC vs. HC	4 SCLC, 8 SCC, 10 ADC, 12 HC	PCA	98.6 % acc.	[246]
	SCLC vs. NSCLC, SCC vs. ADC		DFA	Acc.: 84.5% SCLC vs. NSCLC,77.5% SCC vs. ADC	
**Molecularly capped AuNPs and SWCNT based chemiresistors**	LC with vs. without *EGFR* mutation	19 with *EGFR*, 34 without *EGFR*	DFA	83% acc., 79% sens., 85% spec.	[228]
**Cyranose 320, Tor Vergata, Common In-vent, Owlstone Lonestar**	Clinically stable vs. unstable episodes of asthma	22 partly controlled persistent asthma	PCA	Correct classification: 95% baseline vs. loss of control, 86% loss of control vs. recovery Owlstone Lonestar the most prominent	[90]
**Cyranose 320**	Asthma inflammatory phenotypes	24 EOS., 10 NEUTR., 18 PAUC.	PCA	Acc., sens., spec.: 73%, 60%, 79% EOS. vs. NEUTR., 74%, 55%, 87% EOS vs. PAUC., 89%, 94%, 80% NEUTR. vs. PAUC.	[247]
	Uncontrolled asthma-like symptoms	Training set: 65 cluster 1, 22 cluster 2, 34 cluster 3	one-way ANOVA, Kruskal-Wallis	Significant differences concerning chest tightness during exercise, dyspnea and gender	[248]
	HC and controlled vs. partly controlled and uncontrolled asthma	10 HC, 9 controlled, 7 partly, 12 uncontrolled	PCA, radar plot	Good predictive ability Cross-validated AUC 0.80, 79% sens., 84% spec.	[119]
**Organically-coated AuNPs and SWCNT-based chemiresistor**	BC subtypes	12 LuminalA, 42 LuminalB, 12 Triple Negative, 16 HER2+, 14 HER2 equivocal	DFA	Acc., sens., spec.: LuminalA vs. others 81.3–87.7%, 75–87.5%, 82.1–87.5% LuminalB vs. others78.1–86.3%, 83.3–85.3%, 74.1–87.2% HER2+ vs. others 81.3–82.4%, 81.3–91%, 80.7–81.3% Triple neg, vs. others 82.9–90.3%, 83.3–93.3%, 82.9–89.4% Luminal vs. non-Luminal 70.8–87.7%, 70.4–88.1%, 71.4–87.1% LuminalA vs. LuminalB 85.7–94%, 75–91.7%, 88.2–95.2% HER2 status/luminal 85.7–100%, 85.7–100%, 83.3–100% HER2 status/non-luminal 90.9%, 90.9%, 90.9%	[238]
**Disease staging**					
**Tor Vergata e-Nose**	LC Stage I vs. II/III/IV	40 stage I, 18 stage II, 6 III/IV	PLS-DA	Sens.: stage I 90% vs. stage II-IV 57% (+ metabolic diseases), stage I 96% vs. stage II-IV 60% (LC only)	[133]
**24 colorants**	LC Stage I/II vs. LC stage III/IV	41 SCLC, 42 NSCLC	LPM	79.3 % acc. (moderate)	[127]
**11 sensor-array (MOS, electrochemical, hot wire and catalytic)**	LC Stage III vs. IV	44 stage II, 46 stage IV	PCA-SVM, KPCA-SVM, PCA-XGBoost, KPCA-XGBoost	70.42–82.42% acc., 45–81% sens., 79–95.5% spec.	[227]
**Organically-coated AuNPs and SWCNTs based chemiresistor**	OLGIM stages	155 OLGIM 0, 136 OLGIM I-II, 34 OLGIM III-IV, 7 Dysplasia	DFA	Acc., sens., spec.: 0-II vs. III-IV and dysplasia 61%, 83%, 60%, 0 vs. I-II 43%, 45%, 41%, 0 vs. III-IV 66%, 90%, 61%, 0 vs. I-IV 50%, 50%, 50%, I-II vs. III-IV 64%, 80%, 60%	[239]
	GCa I-II vs. III-IV	17 GCa I-II, 18 GCa III-IV	DFA	89% sens., 94% spec.	[240]
**Molecularly modified SiNW FET**	LC staging (I-II vs. III-IV)	34 early stage, 110 advanced stage	DFA, ANN	81% acc., 34.5%sens.,95% spec.	[43]
	GCa staging (I-II vs. III-IV)			86.5% correct classification, 84.6 early stage, 87.5 advanced	
**Cyranose 320**	Bronchial/Laryngeal in situ vs. advanced	bronchial: 10 in situ, 10 advanced, laryngeal: 12 in situ, 10 advanced	JMP Pro	21% misclassification rate, 82% sens., 75% spec.	[232]
**BIONOTE**	Liver cirrhosis (A, B, C Child–Pugh)	NA	PLS-DA	Successful discrimination	[244]

Acc.: accuracy, AD: Alzheimer’s disease, ADC: adenocarcinoma, AUC: Area Under the Receiver Operating Characteristic Curve, ANN: artificial neural network, BC: breast cancer, BBT: Breast benign tumor, BPN: benign pulmonary nodules, CAP: canonical analysis of principal coordinates, CD: Crohn’s Disease, CDA: canonical discriminant analysis, CF: cystic fibrosis, CLD: chronic liver disease, COP: cryptogenic organizing pneumonia, COPD: chronic obstructive pulmonary disease, CTD-ILD: connective-tissue diseases-associated ILD, DCIS: ductal carcinoma in situ, DFA: Discriminant Function Analysis, DIP: desquamative interstitial pneumonia, EOS: eosinophilic, GCa: gastric cancer, GOLD: Global Initiative for Obstructive Lung Disease, HBC: hexabenzocoronene, HCA: Hierarchical Cluster Analysis, HER2: Human epidermal growth factor receptor 2, HNSCC: Head and neck squamous cell carcinoma, HP: hypersensitivity pneumonitis, IBD: Inflammatory Bowel Diseases, IBS: Irritable Bowel Syndrome, ILDs: interstitial lung diseases, IPAF: interstitial pneumonia with autoimmune features, IPF: idiopathic pulmonary fibrosis, k-NN: k-Nearest Neighbors, KPCA-XGBoost: kernel principal component analysis—extreme gradient boosting, LC: Lung cancer, LRA: logistic regression analysis, LPM: Logistic prediction model, MANOVA: multivariate analysis of variance, MOS: metal oxide semiconductor, NC-CLD: non-cirrhotic CLD, NEUTR: neutrophilic, NS: non-smokers, NSCLC: non-small cell lung cancer, NSIP: non-specific interstitial pneumonia, NWs: nanowires, OC: ovarian cancer, OLGIM: operative link on gastric intestinal metaplasia, PAH: Polycyclic aromatic hydrocarbons, PAUC: paucigranulocytic, PCA: Principal Component Analysis, PD: Parkinson’s disease, PLS-DA: Partial Least Square Discriminant Analysis, PUD: peptic ulcer disease, RB-ILD: respiratory bronchiolitis-associated ILD, S: smokers, SCC: squamous cell carcinoma, SCLC: small cell lung cancer, Sens.: sensitivity, Spec.: specificity, SVM: Support Vector Machine, TB: Tuberculosis, UC: Ulcerative Colitis, uILD: unclassifiable ILD, UV: ultra-violet.

Concerning LC histology and staging with sensing devices, promising studies have been reported in the literature. The discrimination of NSCLC subtypes ADC and SCC has been permitted using Tor Vergata e-Nose with an accuracy of 75%, by applying endoscopic breath sampling [132] as well as by using a colorimetric sensor-array of 24 elements developed by Mazzone et al. ultimately achieving an accuracy of 86.4% [127]. SCLC and NSCLC differentiation and LC staging (I/II vs. III/IV) were also examined by Mazzone et al. though with moderate accuracies [127]. A 6-sensor-array based on UV-irradiated (394 nm) pristine or metal-doped WO_3_NWs (Table 4) differentiated effectively not only ADC from SCC, but also between SCLC and NSCLC with 77.5% and 84.5% accuracy values, respectively (Figure 15) [246]. In another study aiming at the discrimination of LC patients from HC while taking into account the existence of metabolic comorbidities, Tor Vergata e-Nose exhibited far higher sensitivity for stage I LC in comparison to the rest of stages, either in the presence or absence of metabolic diseases (Table 4) [133]. LC staging was recently attempted by Liu et al. along with COPD discrimination as mentioned above, with stage III LC being effectively discriminated from stage IV with an accuracy higher than 80%, using KPCA-XGBoost [227]. Haick’s team has achieved LC staging with an accuracy of 81% and with low sensitivity, using a molecularly modified Si NW FET (Table 4) [43].

As in the case of breath analysis with analytical methods, precise diagnosis of lung diseases other than LC via sensing devices is an extensive field of research. The effective discrimination of COPD and asthma has been reported in the literature by Fens at al. using Cyranose 320 and taking into consideration smoking habits, leading to high cross-validated accuracy values (Table 4) [229,230]. More recently, asthma and CF discrimination was also reported for pediatric population using AeoNose and with high accuracy values, excluding the confounding factors of diet, exercise, comorbidities and inhaled drugs [235]. Concerning ILDs, Krauss et al. used the AeoNose in an attempt to differentiate between ILDs subgroups (Table 4), with moderate accuracy, as well as between ILDs cryptogenic organizing pneumonia and connective-tissue diseases-associated ILD from COPD patients with good sensitivity and specificity [234]. COPD and IPF differentiation has been recently investigated by Dragonieri et al., with a high accuracy of 80%, verified by external validation using Cyranose 320 [231]. In contrast to Krauss et al., Moor’s group achieved to reliably discriminate patients suffering from different ILDs by using SpiroNose as well as greater cohorts of ILD-patients (Table 4) [207]; the group demonstrated the applicability of e-Noses in ILDs differential diagnosis and specifically in IPF discrimination from non-IPF patients with high accuracies (91%) [207].

Disease phenotyping using sensing devices seems to be also feasible. Plaza et al. achieved differentiation between the three inflammatory phenotypes of asthma with high accuracy values (Table 4), with the participants’ phenotypes being characterized by differential leukocyte counts in induced sputum [247] while asthma-control assessment has been also reported. Brinkman et al. used 4 different e-Noses in order to discriminate between stable and unstable periods, comparing baseline (control) vs. loss of control and loss of control vs. recovery breath samples, with the Owlstone Lonestar being the most prominent concerning the discrimination of unstable periods (Table 4) [90]. More recently, Moreira et al. demonstrated the ability of Cyranose 320 to discriminate the uncontrolled asthma-like symptoms, using 3 different groups of asthmatic or suspicious of asthma participants divided by unsupervised hierarchical clustering [248]. The division of participants was based on asthma, lung function, symptoms of the last month, age, and food/drink intake 2 h before breath sampling [248]. In another recent study, the same e-Nose was used for the effective discrimination of HC and asymptomatic-controlled asthmatic children from the symptomatic partly-controlled and uncontrolled asthmatic children, after assessing the discriminatory ability of subsets of the 32 sensors of Cyranose 320 for the six different possible combinations of the 4 studied groups; increased feasibility and modest to good diagnostic accuracy values were obtained [119]. Cyranose 320 has been also used for COPD phenotyping permitting (especially in the case of GOLD stage I) the detection of activation of inflammatory cells, indicating increased inflammatory activity in mild rather than severe COPD [120].

### 4.2. Cancers

Discrimination between different cancer types and cancer stages/histologies as well as between malignant and benign tumors (additionally to LC which has been mentioned earlier) using breath analysis of VOCs is also a hot research topic. Analytical techniques have been used for such applications. Phillips at al., for example, detected 5 VOCs and were able to differentiate BC patients from patients with abnormal mammograms and negative biopsies, with 93.8% sensitivity and 84.6% specificity [249]. Haick’s group identified 21 exhaled VOCs that were significantly different between HC and patients suffering from breast benign tumors (BBT), ductal carcinoma in situ (DCIS, early-stage BC) and BC; a potentially cancer-related set of 14 VOCs that were significantly different between malignant and non-malignant patients was also identified in this study thus permitting group differentiation with 78% sensitivity and 72% accuracy [238]. In another study by the same group, the detection of GCa and the presence/absence and risk level of precancerous lesions was attempted using GC-MS so as to identify 8 VOCs statistically different between GCa and operative link on gastric intestinal metaplasia (OLGIM) groups (e.g., GCa vs. OLGIM 0-IV, GCa vs. OLGIM 0-II, GCa vs. OLGIM 0), as well as between GCa and peptic ulcer disease (PUD) and OLGIM 0-IV and PUD (*p*-values < 0.017) [239]. Those 8 VOCs, in different combinations, are considered to correspond to the breathprints of OLGIM groups [239].

Remarkably, respective applications of sensing devices have been extensively investigated for various cancer types. Haick’s group has used NA-NOSE in order to discriminate between subjects with BC, benign breast conditions or normal mammographs, achieving increased sensitivity and specificity values for the BBT patients in comparison to the other 2 groups [116]. In another study by the same group a chemiresistor based on organically-coated Au NPs and SWCNTs was successfully used for the differentiation of BC from BBT and HC, BBT only or DCIS only, as well as for the differentiation of different molecular BC sub-groups as presented on Table 4. Larger studies are necessary, though for significant statistical results and more information to be obtained [238]. Very recently, differentiation between BC and LC has been achieved as well by Rodriguez et al. with a correct classification of 93.05% [201]. Concerning GCa, the differentiation of GCa and OLGIM groups as well as between different OLGIM stages (e.g., OLGIM 0 vs. I–II, 0 vs. I–IV, I–II vs. III–IV) has been attempted by Haick’s group using the same type of chemiresistor-array and leading to high validated accuracy values in some cases (Table 4) [239]. In another promising study, Haick et al. used a nanomaterial-based chemiresistor (Table 4) that permitted the successful discrimination of Gca from benign conditions along with Gca staging (early vs. late stages) [240]. Gca differentiation from gastric ulcer patients has been achieved by Daniel et al. with a great classification ability, using an array of commercial MOS gas sensors and various ANN types [241]. Remarkably, a molecularly modified Si NW FET, developed by Haick’s group, has permitted Gca staging with an accuracy of 87% as well as Gca differentiation from LC with an accuracy of 92% [43]. More recently, discrimination of ovarian cancer (OC) from women with benign tumors and HC was achieved by Raspagliesi et al. with great classification performance, both in the case of strict and most probable prediction, using again MOS sensors [242]. Notably in class prediction application, 4/23 early-stage OC patients were misclassified as benign/HC along with 2/14 OC patients with tumor size < 3 cm in cross validation phase, while in prediction phase only 1/9 early-stage patients were misdiagnosed [242]. Another common cancer type, i.e., head and neck cancer (HNC), has been studied for differential diagnosis with sensing devices. As an example, Hooren et al. attempted to discriminate HNC from LC with a high accuracy of 93% analyzing patients’ exhaled breath with AeoNose, excluding cutaneous tumors and salivary glands malignancies [236]. HNC differentiation from colon and bladder cancer with AeoNose was also reported, by the same group along with the discrimination between bladder and colon cancer, demonstrating the discriminant ability of the e-Nose for those cancer types after double cross-validation [237]. More recently, Cyranose 320 was used for the differentiation of advanced bronchial (LC) and laryngeal (HNC) SCC, as well as for the discrimination of advanced and in situ stages of bronchial and laryngeal SCC, leading to successful classification of the groups (Table 4) [232].

### 4.3. Liver, Renal, and Intestinal Diseases

Liver cirrhosis, chronic hepatitis [244], CKD [113], and inflammatory bowel diseases (IBD) comprise common liver, kidney, and intestinal diseases, respectively. As far as CKD is concerned, it is characterized by gradual loss of kidney function within months or years, while different treatment is demanded depending on disease stage (stages I–V) [113]. Similarly, in the case of chronic liver disease (CLD), disease staging is of great importance; following CLD diagnosis, using invasive biopsy, liver function assessment is conducted biochemically [244]. On the other hand, early stage and precise IBD and IBS diagnosis, as well as the invasive diagnostic methods followed for IBD, comprise challenging issues [111]. Consequently, precise diagnosis and staging of liver, kidney and intestinal diseases are particularly important and have been attempted using breath analysis with sensing devices. Pennazza et al. for instance used BIONOTE e-nose to successfully differentiate not only liver cirrhosis and CLD from non-cirrhotic CLD (chronic hepatitis) but also liver cirrhosis stages by taking into account smoking habits and potential comorbidities (e.g., diabetes, lung, and heart diseases) [244]. Concerning renal diseases, Haick’s group achieved CKD staging using organically functionalized Au NPs-based chemiresistors and SVM [113]. The classification of CKD stage IV in relation to stage V was permitted by 2 or 3 sensors with an accuracy of 85%, a sensitivity of 75% and specificity of 92% while only one sensor allowed for the discrimination of early and advanced stages with 76% accuracy, 75% sensitivity, and 77% specificity [113]. The discrimination of CKD from other diseases has been also attempted. Specifically, discrimination of CKD, diabetes, and HC with high or low creatinine has been attempted with success using an array of commercial (MQ) sensors along with different classification methods, with SVM and PCA leading to good group classification [245]. Remarkably, pre-concentration or dehumidification were not needed for clear classification to be accomplished [245]. The effective discrimination between the intestinal diseases IBD and IBS has been also reported along with further differentiation of the IBD into ulcerative colitis (UC) and Crohn’s disease (CD), using not only artificial but also real-breath samples and a MCNPs-based chemiresistor (Table 4) [111]. The higher accuracy values observed when using artificial samples is expected and attributed to the standard concentration of VOCs, contrary to the variable concentration of VOCs in breath [111].

### 4.4. Neurodegenerative Diseases

Neurodegenerative diseases are characterized by gradually augmented occurrence, as a direct consequence of the increased lifespan of human population [250], with Alzheimer (AD) and Parkinson (PD) being the most frequent [114]. Concerning AD, early disease detection is of great importance for preventing, decelerating and terminating the disease [250] while diagnosis for both diseases is based on the assessment of clinical symptoms [114]. Remarkably, the analysis of exhaled VOCs has been investigated for precise AD diagnosis as well as for differential diagnosis between AD and PD [114,250]. Recently, Tiele et al. attempted to discriminate mild cognitive impairment due to AD (MCI) from AD, using GC-MS, achieving 60% sensitivity and 84% specificity along with the detection of 6 potential discriminant VOCs [250]. Haick’s group, on the other hand, achieved the discrimination of AD and PD as well as an overall discrimination of AD, PD, and HC, using an array of 20 nanomaterial-based chemiresistors and GC-MS (Table 4) [114]. In a similar study, the ability of IMS and Cyranose 320 to differentiate AD, PD, and HC was demonstrated, achieving a high overall discriminant capability (Table 4) [233]. IMS analysis revealed five VOCs significantly different between the groups [233]. The discrimination of the same groups of subjects has been also attempted using different arrays of MOS sensors (TGS, MICS) with one of the combinations (8 MOS sensors) demonstrating the best discriminant ability [243].

## 5. Conclusions and Future Perspectives

Exhaled breath analysis, especially using selective or cross-reactive sensors, comprises a non-invasive method that holds a great promise for application in early-stage and differential diagnosis of not only respiratory but also systemic diseases. The aim of this review was to present the main categories of nanomaterials and sensors that have been used up to now in exhaled breath analysis for disease diagnosis as well as to demonstrate the applicability of breath analysis in differential diagnosis, phenotyping, and staging of several types of diseases, especially via the use of cross-reactive sensing devices.

The progressive development of novel nanomaterials offers a great opportunity to develop more effective sensing elements, both for selective and cross-reactive sensors and especially for point-of-care diagnosis, treatment monitoring and population screening. However, fundamental challenges in this novel research field inhibit the application of breath analysis in clinical practice and should therefore be addressed. Concerning analytical techniques used for exhaled VOCs identification, the use of bulky, expensive, and complex analytical devices is limited in hospitals while their incorporation in portable point-of-care systems is still unattainable [251]. In addition, the validity of breath analysis results is of major concern since the trace levels of exhaled VOCs affect the analysis accuracy [252]. At the same time the lack of clear breath sampling protocols [252], e.g., breath collection [251] and breath storage, could potentially change sample composition [251] and therefore emerge as important challenges. Sample composition can be also affected by confounding factors, i.e., age, gender, place of living, habits, and nutrition. In the case of sensors exhalation rate, a hardly controlled parameter, may also play a confusing role hence complicating the procedure [251].

For those limitations to be overcome, breath analysis research should focus on sampling and procedure protocols standardization [118,251] and system improvement towards technical/physiological/pathophysiological confounders [118], in order to determine the endogenous VOCs and to define valid exhaled patterns of biomarkers. Towards this direction, standard correlations between blood and breath VOC concentrations could be established [251]. Furthermore, the development of portable/wearable and low-cost nanomaterial-based sensors that are resistant to humidity [251] and serve the clinical needs (i.e., selectivity for disease-specific VOCs and inorganic gases, small recovery time for population screening), along with the optimization of sensor training and validation by using different subject-groups comprise critical steps for the development of sensors applicable in clinical diagnosis [118]. Notably, the ability of new sensing systems to discriminate different diseases, to achieve precise early diagnosis of diseases with similar symptoms and underlying mechanisms, is a major concern [200] that has only recently been considered.

In order to meet all of the aforementioned goals, interdisciplinary research and co-operation is an essential prerequisite [251]. Breath analysis poses as a powerful and promising diagnostic tool that could eventually be used in clinical practice or in portable and compact health monitoring systems, provided it meets most of the existing challenges.

## Figures and Tables

**Figure 1 sensors-22-01238-f001:**
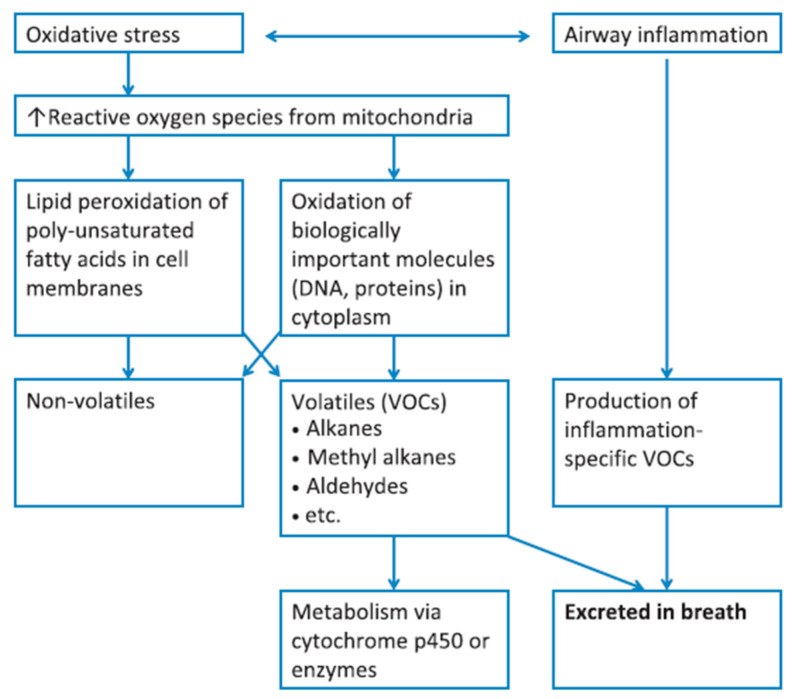
Diagram summarizing the correlation of VOCs present in the exhaled breath, with oxidative stress and inflammatory conditions; metabolic breakdown of larger molecules leads to the formation of exhaled VOCs. Reprinted with permission from ref. [17]. Copyright © 2012 John Wiley & Sons Ltd.

**Figure 2 sensors-22-01238-f002:**
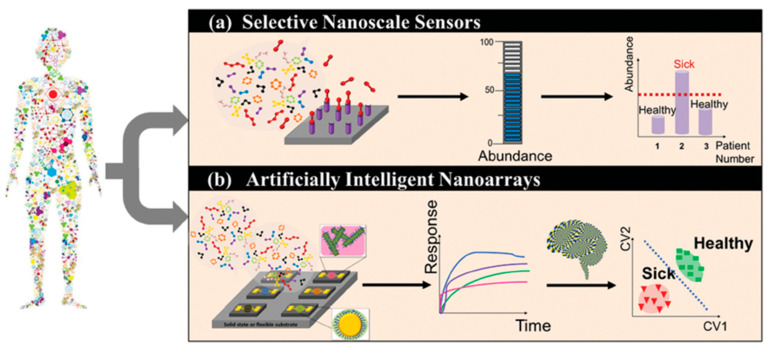
Schematic representation of the working principle of selective sensors and artificially intelligent cross-reactive sensor arrays. Selective sensors contain highly selective elements in order to detect a specific gas-analyte in the presence of a composite gas-mixture. Cross-reactive arrays feature sensors that are sensitive to the majority of the gases present in the gas-mixture. In any case, detecting analyte concentration above a critical value leads to the differentiation between sick and healthy subjects. The response of gas-sensing arrays can be then processed by employing artificial intelligence, machine-learning, and pattern recognition techniques. Reprinted with permission from Ref. [6] Copyright © 2015 Wiley-VCH Verlag GmbH & Co. KGaA, Weinheim.

**Figure 3 sensors-22-01238-f003:**
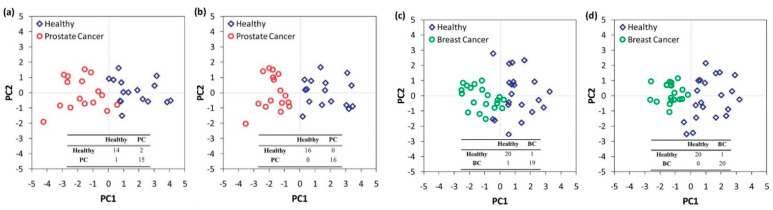
Statistical analysis of the response of a nanomaterial-based, cross-reactive chemiresistor for real-world samples of sick and healthy subjects. The use of PCA permits the differentiation of the groups. Notably, relative humidity compensation reduces the dispersion of different clusters thereby improving the discrimination between healthy and sick subjects. Representative 2D breath-analysis PCA plots for prostate cancer diagnosis: (**a**) without relative humidity compensation; (**b**) with relative humidity compensation. PCA plots for breast cancer diagnosis: (**c**) without relative humidity compensation; (**d**) with relative humidity compensation. Adapted with permission from Ref. [42] Copyright © 2012, American Chemical Society.

**Figure 4 sensors-22-01238-f004:**
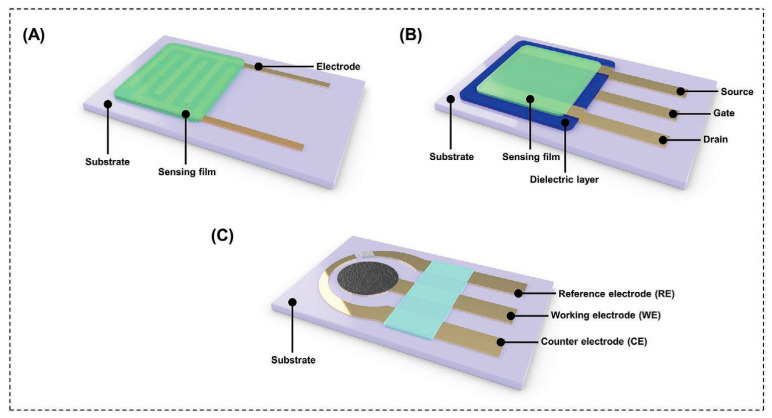
Schematic representation of (**A**) a chemiresistor under a constant bias and with an overlying metallic or semiconducting sensing layer which acts as the sensitive gas-sensing layer; (**B**) a field-effect transistor where the conductivity of the channel is sensitive to gas-analytes exposure; (**C**) an electrochemical sensor (potentiometric, amperometric, or conductometric) composed of a working (sensing) electrode on which the analyte reacts (redox reaction), a counter electrode (with respect to which electrical signal is measured) and a reference electrode of “reference” potential. Reprinted with permission from Ref. [13] Copyright © 2021 Wiley-VCH GmbH.

**Figure 5 sensors-22-01238-f005:**
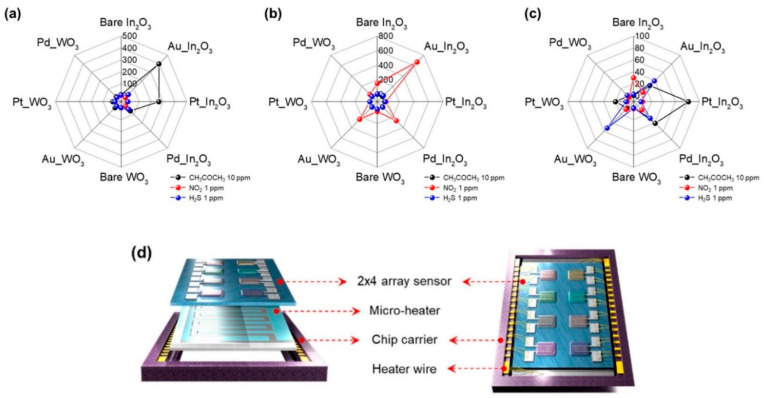
Polar plot of a 2 × 4 sensor array response for (**a**) 10 ppm of acetone; (**b**) 1 ppm of NO_2_ and; (**c**) 1 ppm of H_2_S and operation temperatures 300, 150, and 250 °C; (**d**) schematic representation of the parts of the sensor (2 × 4 sensor array, back heater, chip carrier, and Au wires). The array shows good recovery and repeatability as well as high performance at sub ppb level, facilitating discrimination between the three biomarkers. Adapted with permission from Ref. [99]. Copyright © 2021 by the authors. Licensee MDPI, Basel, Switzerland.

**Figure 6 sensors-22-01238-f006:**
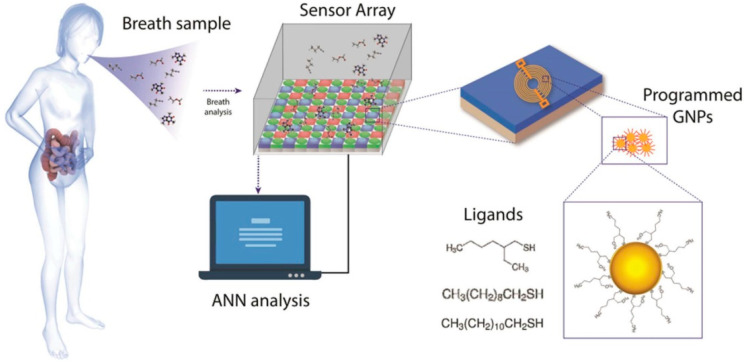
Schematic representation of a MCNPs-based chemiresistor, programmed according to the VOCs identified as inflammatory bowel and irritable bowel syndrome disease biomarkers. The sensor was exposed to simulated, HC and patient breath samples. Time-dependent and reversible shift in the sensor’s resistance is associated with the MCNPs–VOCs interactions. The variability of molecular ligands leads to varying sensing responses; pattern recognition methods such as ANN, permit the development of effective diagnostic classifiers. Reprinted with permission from ref. [111]. Copyright © 2016 WILEY-VCH Verlag GmbH & Co. KgaA, Weinheim.

**Figure 7 sensors-22-01238-f007:**
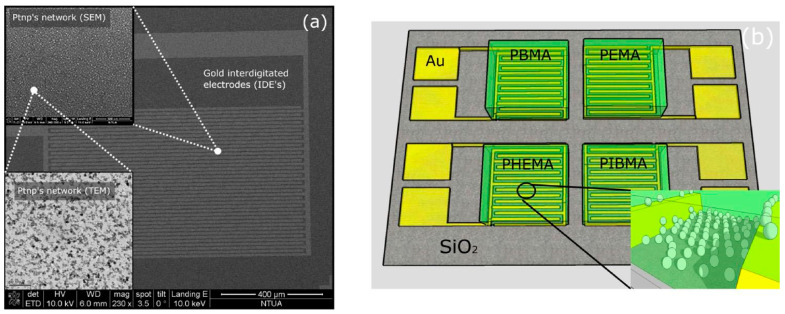
(**a**) SEM image of an array of polymer-coated PtNPs-based chemiresistors, developed for pesticide detection, TEM image of PtNPs layer, with surface coverage 46% and mean nanoparticle diameter of 5 nm; (**b**) schematic representation of the sensor array. Different polymer susceptibility towards water/pesticide vapors leads to a gas-sensing array that is capable of identifying each of the gas-analytes. Reprinted with permission from Ref. [41] Copyright © 2018 Elsevier Ltd.

**Figure 8 sensors-22-01238-f008:**
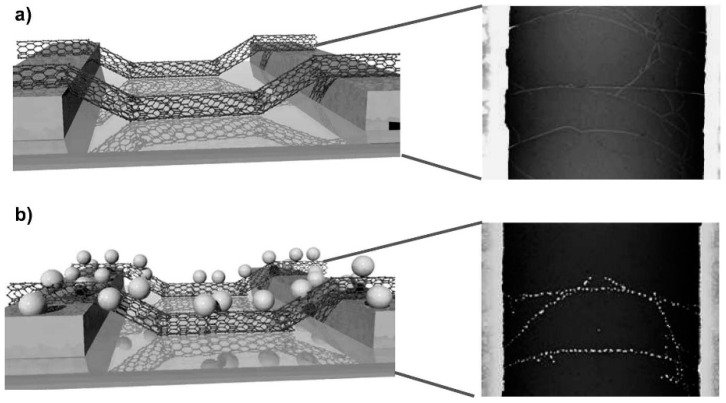
Schematic representation and SEM images of (**a**) carboxylated SWCNTs aligned between Au electrodes and (**b**) Au-decorated SWCNTs, as sensing films for gas sensors (FETs), for enhanced H_2_S sensing. Au NPs play a crucial role in device-performance by modulating the mobility after the gaseous-molecule interactions. Reprinted with permission from Ref. [156]. Copyright © 2011 Wiley-VCH Verlag GmbH & Co. KgaA, Weinheim.

**Figure 9 sensors-22-01238-f009:**
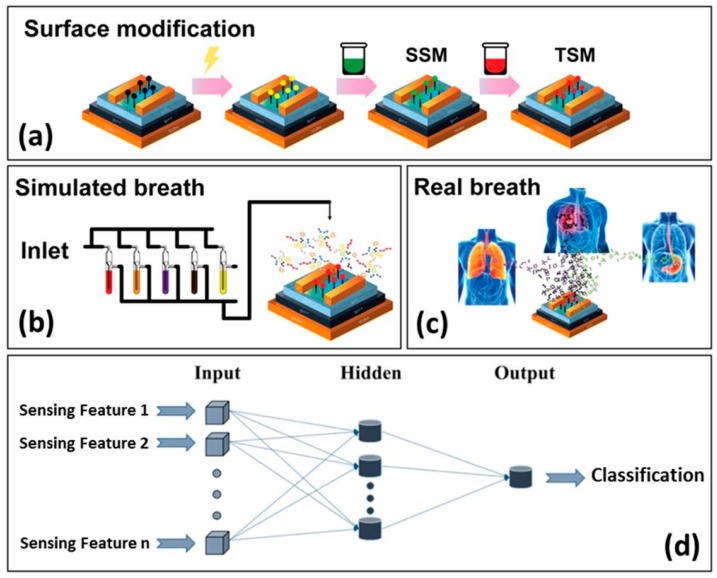
Schematic of (**a**) surface modification of SiNW FET sensors (SSM, single-step modification; TSM, two-step modification); (**b**) sensor exposure to synthetic samples of potential organic volatile biomarkers for each disease (asthma/COPD, LC, GCa); (**c**) sensor exposure to real breath samples of patients with selected diseases, in comparison to HC; (**d**) ANN analysis representation. Increased accuracies were obtained for the discrimination of GCa vs. LC and LC vs. asthma and COPD patients. Reprinted with permission from Ref. [43]. Copyright © 2016, American Chemical Society.

**Figure 10 sensors-22-01238-f010:**
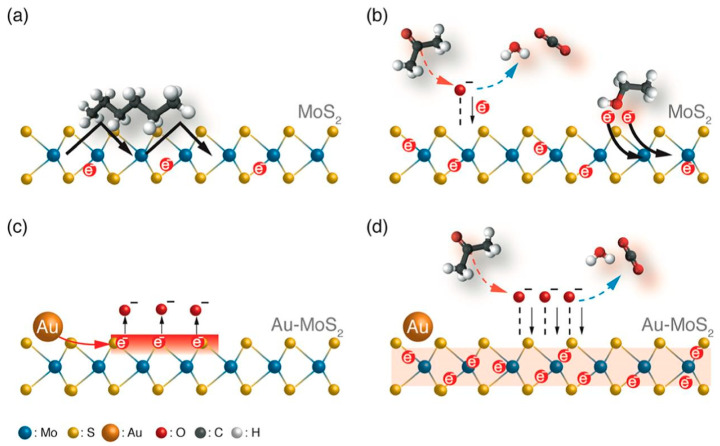
Schematic representation of Au-MoS_2_ nano-flakes sensing mechanisms for (**a**) hydrocarbons using MoS_2_, leading to small signal alternation, due to dipole scattering through the electron clouds of –CH_2_– groups; (**b**) oxygen-based VOCs using MoS_2_, which donate electrons to the MoS_2_ (**left**) and are oxidized by the adsorbed oxygen species, releasing electrons to MoS_2_ (**right**); (**c**) oxygen-based VOCs using MoS_2_ decorated with Au NPs, which increase electron concentration and, thus, oxygen species adsorption on MoS_2_; (**d**) returning more electrons upon interaction with the oxygen-based VOCs. Reprinted with permission from ref. [69]. Copyright © 2019, American Chemical Society.

**Figure 11 sensors-22-01238-f011:**
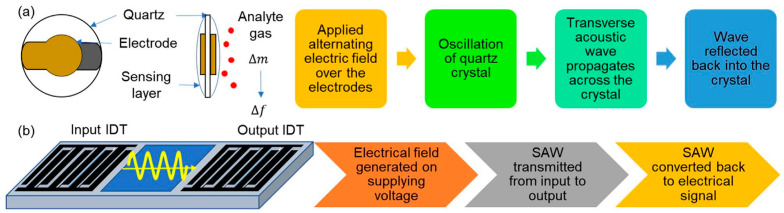
Representation of the sensing mechanism of gas detection using (**a**) a QCM sensor where the oscillation of the quartz crystal substrate and propagation of the transverse acoustic wave through the quartz substrate are caused by the alternating electric field applied over the electrodes; sensing layer-gas interactions change the mass on the substrate and hence wave amplitude and velocity, leading to a resonance frequency shift (Δm → Δf); (**b**) SAW sensors where a surface wave confined within one acoustic wavelength of the surface of the piezoelectric material is induced by an input RF-voltage applied across the interdigitated transmitter (IDT); mechanical energy is transformed back into radio frequency as an output when the SAW reaches the receiving IDTs. Analyte adsorption on the piezoelectric coating induces mass variations and ultimately a shift in frequency. Reprinted with permission from Ref. [172]. Copyright © 2021 Elsevier B.V. All rights reserved.

**Figure 12 sensors-22-01238-f012:**
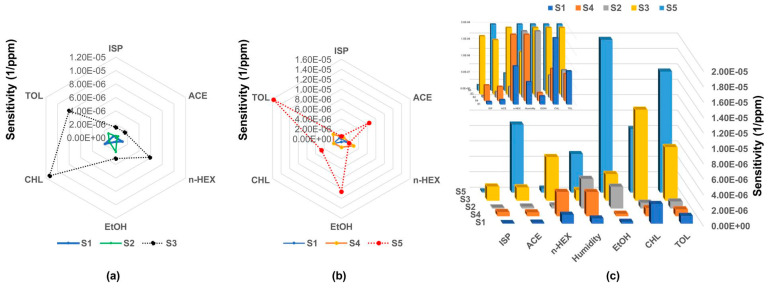
Sensitivity values (1/ppm) of (**a**) sensors S1–S3, (**b**) sensor S1–S4–S5 in radar plots, and (**c**) sensors S1–S5 in bar chart representation, for six different analytes under 0% RH provided with responses to 100% humidity. Inserted graph: magnified view of data below 2 × 10^9^ 1/ppm. (S1: SH-Calix[4]arene, S2: AuNRs, S3: Calix[4]arene modified AuNRs, S4: AgNCs, and S5: Calix[4]arene modified AgNCs, ISP: isoprene, ACE: acetone, n-HEX: n-hexane, EtOH: ethanol, CHL: chloroform, TOL: toluene). Calix[4]arene modification (S3, S5) increased the sensitivity, under 0% of humidity, especially for TOL and CHL, while thiol terminated calix[4]arene (S1) exhibited increased response towards CHL. S5 exhibited the highest responsiveness towards TOL among other VOCs and the highest among all sensors (π–π interactions leading to host–guest complexes). Reprinted with permission from Ref. [49]. Copyright © 2021 Elsevier B.V. All rights reserved.

**Figure 13 sensors-22-01238-f013:**
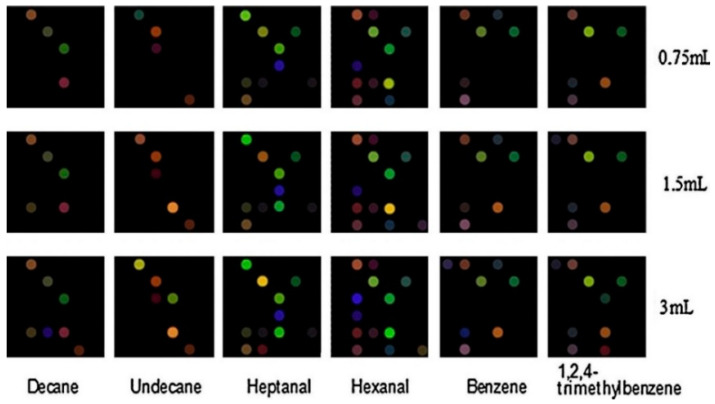
Detailed color difference maps of six VOCs at three volumes (0.75, 1.5, and 3 mL saturated vapor, respectively) saturated analyte vapor at 20 °C, acquired by metalloporphyrin-AuNRs and dyes-based optical chemical sensor. Metalloporphyrin-AuNRs exhibited increased responsiveness and high sensitivity and selectivity due to their high affinity. Reprinted with permission from Ref. [141] Copyright © 2014 Elsevier B.V. All rights reserved.

**Figure 14 sensors-22-01238-f014:**
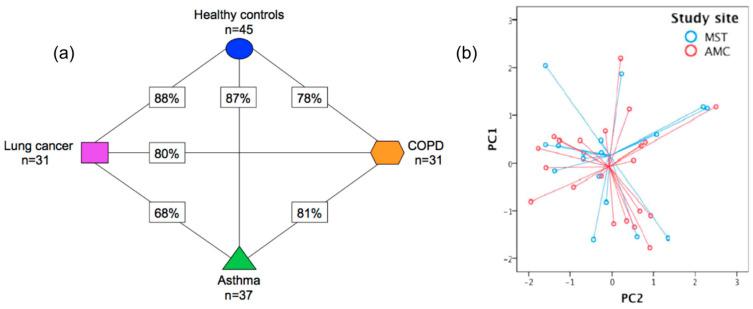
(**a**) Presentation of the cross-validation percentages of the differentiation of asthma, COPD, LC patients, and HC, using SpiroNose; (**b**) PCA plot of breathprints collected from asthmatic patients at the Academic Medical Center (AMC), Amsterdam and Medical Spectrum Twente (MST), Enschede, for which no significant differentiation is observed (*p* = 0.892). Adapted with permission from Ref. [91]. Copyright © 2015 IOP Publishing Ltd.

**Figure 15 sensors-22-01238-f015:**
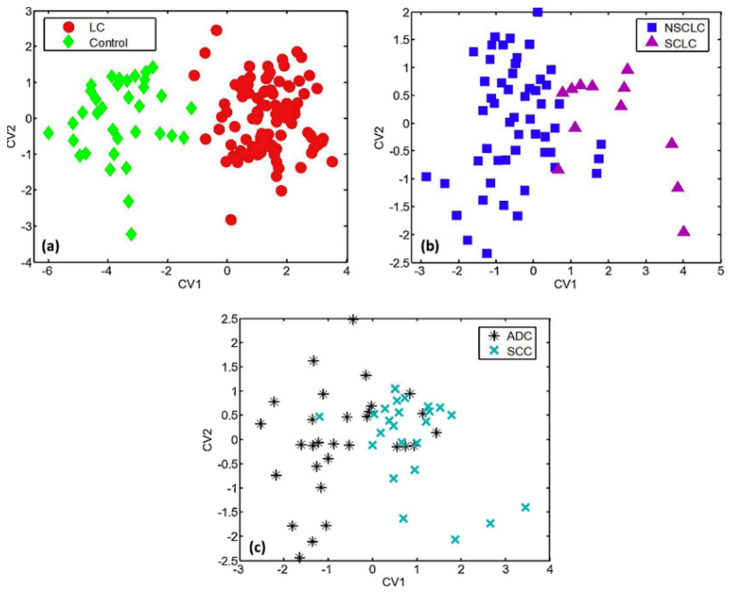
DFA plots representing the discrimination of (**a**) LC patients from HC; (**b**) SCLC from NSCLC patients; and (**c**) SCC from ADC patients, using a 6-sensor array of UV-irradiated (394 nm) pristine or metal-doped WO_3_NWs. The arrays achieved the detection of lung cancer but also the prediction of LC histological subtypes. Reprinted with permission from Ref. [246]. Copyright © 2020 Published by Elsevier B.V.

**Table 1 sensors-22-01238-t001:** Diseases investigated for diagnosis using breath analysis.

Disease Type	Diseases	Ref.
**Respiratory**	Asthma, COPD, obstructive sleep apnea syndrome, pulmonary arterial hypertension, cystic fibrosis	[19]
**Malignant**	Lung, gastric, head and neck, breast, colon, prostate cancer	[15]
**Neurodegenerative**	Alzheimer’s disease, Parkinson’s diseases, multiple sclerosis	[15]
**Metabolic**	Diabetes, hyperglycemia	[12,31]
**Bacterial infections**	Upper respiratory tract infection, *Mycobacterium tuberculosis*, *Pseudomonas*, *Helicobacter pylori* infection	[32,22]
**Viral infections**	SARS-CoV-2	[24,25,26]

COPD: Chronic obstructive pulmonary disease.

**Table 3 sensors-22-01238-t003:** Branded e-Noses and type of technology used in each case.

Electronic Nose	Technology Used	Ref.
**Cyranose 320**	32 carbon black-polymer composite chemiresistors	[117,118]
**Common Invent e-Nose**	MOS sensors	[90,117]
**SpiroNose**	5 sensor arrays, each composed of 4 MOS sensors	[91,117]
**AeoNose**	Micro hotplate MOS	[92,117]
**DiagNose**	12 doped MOS	[93,118]
**BIONOTE e-Nose**	8 QCM sensors using anthocyanin-coated gold electrodes	[117,136]
**Tor Vergata e-Nose**	QCM covered with metalloporphyrins	[117]
**Owlstone Lonestar e-Nose**	Field asymmetric ion mobility spectrometry	[117]

e-Nose: Electronic Nose, MOS: Metal Oxide Semiconductor, QCM: Quartz Crystal Microbalance.

## Data Availability

Not applicable.

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
