# Peer review of "Breath Analysis: A Promising Tool for Disease Diagnosis—The Role of Sensors"

_sensors, 2022, doi:10.3390/s22031238_

Round 1

Reviewer 1 Report

The review is well structured and written. I have basically a few concerns:

  • please revise the English. there are some typos distributed along with all the manuscript.
  • be sure that the quality of all the figures is adequate.
  • the references. I know that is a hue amount of work but please  use the same style for all the references. 

Author Response

Identification of 2 commercial pesticides by a nanoparticle gas-sensing array

Maria Kaloumenou, Evangelos Skotadis, Nefeli Lagopati, Efstathios Efstathopoulos and Dimitris Tsoukalas

Dear editor,

We would like to thank the reviewers for their insightful and constructive comments. In the text below we address all their comments; our changes in the paper are marked using the track-changes feature of MS-word.

Reviewer 1

The review is well structured and written. I have basically a few concerns:

  • please revise the English. there are some typos distributed along with all the manuscript.

We have now carefully revised and edited our manuscript for any typos or mistakes.

  • be sure that the quality of all the figures is adequate.

We have now extracted the best possible figure-files in terms of quality from the original pdf source-files; these files have been uploaded separately as a zipped file. Please note that we were denied access to any of the original, source image-files that have been used with permission from the respective journals. The new Figures have been extracted as vector-graphics from the original pdf source-files and are comparable (in quality) to the ones published in the original papers.

  • the references. I know that is a hue amount of work but please use the same style for all the references.

Thank you for your comment. We have revised our reference list as well as the references’ appearance throughout the manuscript; our references are now uniform and in the sensors-required format.

Reviewer 2 Report

This review paper reported breath analysis on sensors for disease diagnosis. The topic is interesting and valuable. I think this manuscript can be accepted after the following issues are solved.
1, The marks of the figures are not consistant. Some are (a) and the others are (A). Please make them consistant.
2, I noticed that the key part of this manuscript should be VOC, why mentioned a lot of NO2 and H2S. What is the reason for this gases?
3, It seems that  a reference for a figure? However, there should have many references for a theory in sensing. The authors can make more detail information to make the figures better or show more reference papers.
4, The topic is big but the trend of this one should be on machine learning and data processing. The author can list several reference paper to highlight this point.

Author Response

Identification of 2 commercial pesticides by a nanoparticle gas-sensing array

Maria Kaloumenou, Evangelos Skotadis, Nefeli Lagopati, Efstathios Efstathopoulos and Dimitris Tsoukalas

Dear editor,

We would like to thank the reviewers for their insightful and constructive comments. In the text below we address all their comments; our changes in the paper are marked using the track-changes feature of MS-word.

Reviewer 2

This review paper reported breath analysis on sensors for disease diagnosis. The topic is interesting and valuable. I think this manuscript can be accepted after the following issues are solved.

  • The marks of the figures are not consistant. Some are (a) and the others are (A). Please make them consistant.

We have now uploaded new Figures of better quality and in pdf format (images have been uploaded separately as a zipped file). Unfortunately since the figures are published under license from their respective journals we cannot modify them in any way hence the numbering/mark of the original image cannot be tampered. We hope that the improved quality of the pictures compensates for the lack of uniformity.

  • I noticed that the key part of this manuscript should be VOC, why mentioned a lot of NO2 and H2S. What is the reason for this gases?

It is true that the current paper emphasizes on the role of VOCs in breath analysis while the majority of the gas-analytes discussed in the current paper is of organic nature. However, breath analysis is not limited to the contribution of VOCs and there are numerous articles in the literature (that are also reported in our paper) that highlight the importance of inorganic gases. Since the current paper aspires to report on a broad spectrum of breath analysis applications, the authors decided to include breath analysis applications based on inorganic gas-analytes in addition to VOCs, as can be seen in Section 1 “Introduction”: paragraph 2, lines 7-10. 

The following changes have been made in the main body of our manuscript in order to highlight the importance of gas-analytes other than VOCs. Section 1 “Introduction”: paragraph 2, lines 12-13; paragraph 10, line 4. Section 5 “Conclusion and future perspectives”: paragraph 3, line 8. In addition, changes have been made in our abstract in lines 4-8 & 13. Minor changes can be also found throughout our manuscript.

  • It seems that  a reference for a figure? However, there should have many references for a theory in sensing. The authors can make more detail information to make the figures better or show more reference papers

We have revised the Figure captions of our manuscript. Our captions are now more detailed, providing more information and better analysis of the results reported in the references. In addition, the Figures are now of improved quality since they have been extracted as vector-graphics from the original pdf source-files.

  • The topic is big but the trend of this one should be on machine learning and data processing. The author can list several reference paper to highlight this point.

There are numerous reference papers in our review that focus on the role of data processing and machine learning. We have now updated our manuscript so as to include more discussion related to data processing and machine learning.

Changes can be found in Section 1 “Introduction”: paragraph 8, lines 20-24 and in paragraphs 9-11. Also, the importance of data processing and machine learning is highlighted in Section 3 “Types of Nanomaterial-based Sensors in Breath Analysis”: paragraph 47, lines 14-17.

Round 2

Reviewer 2 Report

The authors did very well about the breath gas on the sensing mechanism. I feel that the version looks ok. But one more thing, Figure 14b is not clear, please make it clear.

Author Response

Dear Editor and reviewers, 

let us thank you once more for your insightful and valuable comments and remarks. Below you can find our reply to reviewer 2.

Reviewer 2:

The authors did very well about the breath gas on the sensing mechanism. I feel that the version looks ok. But one more thing, Figure 14b is not clear, please make it clear.

We have reprocessed image 14. The highest resolution possible has been obtained from the source pdf file for both figure 14 (a) and (b). The png file that can be now found inside our word manuscript has been improved by selecting an improved extraction resolution (5 times higher than before). Finally, the new pdf file that can be now found along with the rest of the figures (Figures.zip file) has been also improved in terms of extraction-resolution (again by 5 times).

We hope that the improved figure is adequate since the authors do not have any access to the original figure source-files; without the original files the authors must rely on the available files that can be found in the online version of the respective paper.

On behalf of the authors, I would like to thank you once more for your help and remarks.